evolution, ecology

allometry, anurans, amphibians, morphological evolution, sensory evolution, visual ecology

**Author for correspondence:**
Kate N. Thomas
e-mail: kate.thomas@nhm.ac.uk

# Eye size and investment in frogs and toads correlate with adult habitat, activity pattern and breeding ecology

Kate N. Thomas[1], David J. Gower[1], Rayna C. Bell[2,3], Matthew K. Fujita[4], Ryan K. Schott[2] and Jeffrey W. Streicher[1]

[1]Department of Life Sciences, The Natural History Museum, London SW7 5BD, UK
[2]Department of Vertebrate Zoology, National Museum of Natural History, Smithsonian Institution, Washington, DC 20560-0162, USA
[3]Department of Herpetology, California Academy of Sciences, San Francisco, CA 94118, USA
[4]Department of Biology, Amphibian and Reptile Diversity Research Center, The University of Texas at Arlington, Arlington, TX 76019, USA

KNT, 0000-0003-2712-2481; DJG, 0000-0002-1725-8863; RCB, 0000-0002-0123-8833; RKS, 0000-0002-4015-3955; JWS, 0000-0002-3738-4162

Frogs and toads (Amphibia: Anura) display diverse ecologies and behaviours, which are often correlated with visual capacity in other vertebrates. Additionally, anurans exhibit a broad range of relative eye sizes, which have not previously been linked to ecological factors in this group. We measured relative investment in eye size and corneal size for 220 species of anurans representing all 55 currently recognized families and tested whether they were correlated with six natural history traits hypothesized to be associated with the evolution of eye size. Anuran eye size was significantly correlated with habitat, with notable decreases in eye investment among fossorial, sub-fossorial and aquatic species. Relative eye size was also associated with mating habitat and activity pattern. Compared to other vertebrates, anurans have relatively large eyes for their body size, indicating that vision is probably of high importance. Our study reveals the role that ecology and behaviour may have played in the evolution of anuran visual systems and highlights the usefulness of museum specimens, and importance of broad taxonomic sampling, for interpreting macroecological patterns.

## 1. Introduction

Vision is an important, well-studied sensory system among vertebrates. The size and dimensions of an eye directly affect the optics of the visual system and, subsequently, the amount and quality of visual information that an animal can extract from its environment [1]. Eyes must balance needs for sensitivity (ability to perceive contrast), acuity (ability to perceive spatial detail) and temporal resolution (ability to perceive change over time) [2]. Sensitivity increases when each retinal photoreceptor views a larger solid angle of the visual scene, allowing more photons to reach each detector. Resolution, however, increases when the solid angle sampled by each photoreceptor is decreased, dividing the external visual scene into finer detail [3]. Consequently, an improvement in one aspect of vision often comes at the cost of another, unless the size of the eye is increased.

When an eye is scaled up with constant proportions, acuity increases, while sensitivity to extended visual scenes does not change. This is because acuity is proportional to focal length, while sensitivity is proportional to the ratio of aperture to focal length [2,4]. However, a number of morphological and neural strategies can improve sensitivity at the cost of acuity [5], so a larger eye can improve both sensitivity and/or resolution compared to a smaller eye. Thus,

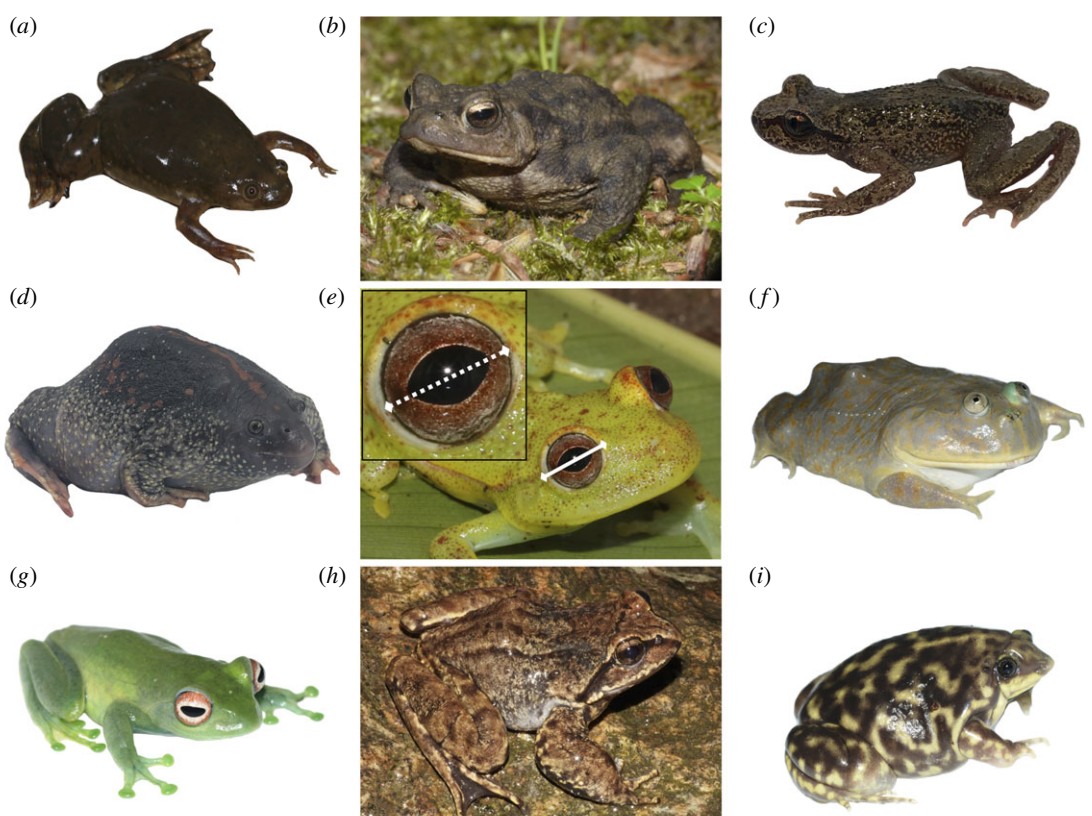

**Figure 1.** Examples of variation in anuran relative eye size across different species and ecologies. (a) *Xenopus laevis* (Pipidae; aquatic); (b) *Bufo bufo* (Bufonidae; ground-dwelling); (c) *Ascaphus truei* (Ascaphidae; semiaquatic); (d) *Rhinophrynus dorsalis* (Rhinophrynidae; fossorial); (e) *Boana punctata* (Hylidae: scansorial) with measurements of eye diameter (solid) and cornea diameter (dashed); (f) *Lepidobatrachus laevis* (Ceratophryidae; aquatic); (g) *Boophis luteus* (Mantellidae; scansorial); (h) *Nanorana liebigii* (Dicroglossidae; semiaquatic); (i) *Hemisus marmoratus* (Hemisotidae; fossorial). Photographs by S. Mahony, J. Streicher, D. Gower, C. Cox and M. Fujita. (Online version in colour.)

understanding eye size across animals with different visual needs is an important part of understanding sensory evolution.

While increasing size improves an eye's ability to collect information from the visual environment, eye size is developmentally and physically constrained. Neural tissues and processing are metabolically expensive even when inactive [6,7], and thus eyes tend to be reduced or lost fairly quickly when no longer useful (e.g. in caves, underground), enabling reallocation of metabolic resources to functional traits [8–10]. Large eyes can also be more vulnerable to damage [11] and may even inhibit locomotion [12]. Thus, relative eye size is expected to reflect a balance of associated costs and benefits that vary depending on the ecology and visual needs of a given species.

Previous studies on birds, fishes, reptiles and mammals have demonstrated that relative eye size in vertebrates typically corresponds to ecological factors. Some fast-moving animals have relatively large eyes supporting high temporal resolution and acuity during transit (e.g. birds of prey [13,14]), though large eyes may also be selected against in fast or far-travelling species (e.g. migratory birds [15]). Activity pattern can also influence eye size, and nocturnal animals often have relatively large eyes and/or eyes with large apertures (pupils) relative to focal lengths to increase sensitivity in low-light conditions (e.g. birds [16]; primates [17], though see [18]; geckos [19]; reef fish [20]); however, the opposite trend has been found in some groups (e.g. snakes [21]). Finally, habitat often correlates with relative eye size because of variation in light levels and propagation among environments (e.g. snakes [21]; geckos [19]; sharks

and rays [22]; mammals [23]). Moreover, early tetrapods showed large increases in relative eye size thought to be adaptive in the transition from vision in water to air and integral to the evolution of terrestriality [24].

The greater than 7100 extant anuran species exhibit stunning phenotypic diversity reflecting over 200 million years of evolution in terrestrial and aquatic habitats (figure 1) [25]. Like other vertebrates, anurans employ vision for activities including intraspecific visual signalling, predator avoidance, prey tracking and discrimination, and habitat selection [26,27]. Despite this, anuran eye size (absolute and relative to body size) is largely unstudied outside of a few families, and potential associations between eye size and ecology are unclear [28]. Additionally, though anurans comprise roughly 10% of all extant vertebrates, eye size and eye–body allometry in frogs and toads have not previously been compared to other vertebrate groups in a phylogenetic framework (though see [29]).

We propose that broad sampling of anuran phylogenetic and ecological diversity potentially relevant to vision may uncover correlations between anuran ecology and relative eye size. Because adult habitat and activity pattern are important drivers of relative eye size in other taxa, we predicted that (i) species active in fossorial (burrowing) and aquatic habitats would show reduced eye investment because they inhabit dark or highly attenuating environments, and that (ii) nocturnal species would invest in large eyes and/or large corneas (approximating maximum pupil diameter) to maximize sensitivity in scotopic (low-light) conditions. Additionally, because many anurans are most active during the breeding season and may be heavily reliant on vision at

this time [30], we predicted that (iii) species breeding in complex sensory habitats (e.g. on vegetation or near fast-flowing water) or (iv) exhibiting sexual dichromatism (potentially related to conspecific signalling [31,32]) would invest in larger eyes for improved visual discrimination during breeding. Finally, because most anurans have a biphasic life history with decoupled larval and adult morphologies and ecologies [33], we predicted that species with (v) free-living larvae and (vi) larvae active in terrestrial or lotic environments (where vision may be crucial to larval survival) would have larger adult eye sizes due to increased larval investment in vision.

In this study, we (i) determine anuran eye–body allometric relationships with comprehensive familial sampling, (ii) test for correlations between eye size and ecology based on our above predictions, and (iii) compare anuran eye size and eye–body scaling to other vertebrate groups.

## 2. Methods

### (a) Sampling and morphological measurement
We selected species to (i) broadly sample taxonomic and ecological diversity, and (ii) match species to a published molecular phylogeny [34]. Our sampling included 100% ($n = 55$) of currently recognized anuran families, and 1–7 specimens for each of 220 species. We examined adult specimens ($n = 640$) from the collections at the Natural History Museum (NHM; London, UK), the North Carolina Museum of Natural Sciences (NCMNS; Raleigh, NC, USA) and the Bombay Natural History Society (BNHS; Mumbai, India). We additionally sampled 67 fresh specimens from 50 species belonging to 17 families for comparison to preserved specimens. Live anurans were obtained through fieldwork in French Guiana (permit RAA: R03-2018-06-12-006) and via collection in the UK or the pet trade (NHM licence NE: WML-OR04).

We collected morphological data with dial-gauge calipers (Helios, Gammertingen, Germany, ±0.1 mm) and digital scales (specimens less than 60 g: CM60-2N, ±0.01 g; specimens greater than 60 g: CM-1KIN, ±1 g, Kern, Balingen, Germany). For each specimen, we measured snout–vent length (SVL; mm), wet mass (g), and external transverse eye diameter (ED; mm) and transverse cornea diameter (CD; mm) for each eye (figure 1e; see [35]). For fresh specimens, we also measured the transverse diameters and axial lengths (AL; mm) of intact, dissected eyes. Mean CD and ED were calculated for each specimen prior to analyses, and we used the cube root of mass (RM) in our models so that isometry with length measurements would occur at a slope of 1.

### (b) Phylogeny
We matched sampled species to a published molecular phylogeny [34,36]. We pruned the 309-species phylogram to species matching our data ($n = 164$) or known close relatives ($n = 56$), and renamed tip labels for close-relative substitutions ($n = 47$) using ape v. 5.3 in R [37]. This was equivalent to adding each close relative into the tree as a polytomous sister taxon and then pruning the unmeasured taxa. We then added remaining species ($n = 9$) as polytomies with their closest known relatives and randomly resolved the polytomies using the multi2di function in ape. All substitutions and polytomies were with congeneric species except for three that were from closely related genera: (i) in Gastrophryninae (*Dermatonotus muelleri* with *Stereocyclops incrassatus*; see [38]), (ii) in Bufonidae (*Altiphrynoides osgoodi* with *Nectophrynoides tornieri*; see [39]), and (iii) in Holoadeninae (*Oreobates quixensis* with *Barycholos pulcher* [40]). For consistency, our analyses and figures follow species names and taxonomy in Amphibian Species of the World [41].

### (c) Ecological classification
We assigned each species in our dataset to a discrete, categorical state for each of six ecological traits (electronic supplementary material, table S1): adult habitat (scansorial, ground-dwelling, subfossorial, fossorial, aquatic, semiaquatic), adult activity period (diurnal, nocturnal, both), mating habitat (lotic water, lentic water, plants, ground), life-history strategy (presence or absence of free-living larvae), larval habitat (lotic water, lentic water, on land, obscured) and sexual dichromatism (present or absent). We used published literature, online natural history resources, field guides and field observations to categorize species ecology (see Supplemental References in the electronic supplementary material). Species that were missing information for a trait were excluded from analyses of that trait.

### (d) Validating specimen measurements
Using museum specimens to quantify eye and body size presents two potential problems: (i) external eye diameter measurements may not be as accurate as measurements on dissected whole eyes, and (ii) fixation and preservation may alter tissue sizes and shapes (e.g. [42]), potentially introducing error into allometric scaling relationships. Ideally, we would avoid these issues by using fresh specimens, but this was not feasible for the taxonomic range of species in our analyses. Instead, we collected additional morphological data from fresh specimens ($n = 67$) to test (i) how well whole, dissected eye diameters correlate with eye diameters measured externally prior to dissection in the same specimens, and (ii) whether eye–body allometry derived from fresh specimens differs from eye–body allometry derived from preserved specimens. See Supplemental Methods in the electronic supplementary material for details.

### (e) Investigating eye to body size allometry in anurans
We fit phylogenetic generalized least-squares (PGLS) regressions with maximum-likelihood estimations of $\lambda$ to log-transformed species means for ED versus SVL, ED versus RM, CD versus ED, CD versus SVL, CD versus RM, and SVL versus RM using caper v. 1.0.1 in R, and used standard checks for model fit (i.e. Q-Q plots, residuals versus fitted values, observed versus fitted values) [43]. Extreme outliers in phylogenetic residuals can overly influence PGLS model fits and parameter estimates [44], so we performed a sensitivity check by iteratively removing species with Studentized residuals exceeding |±3| (see [45,46]) and re-running PGLS models. Removal of phylogenetic outliers had little effect on parameter estimates (electronic supplementary material, table S3), so we used full datasets in final analyses. For comparison to allometric studies that are not phylogenetically corrected, we also fit ordinary least-squares (OLS) regressions using stats v. 3.6.2 [47] and standardized major axis (SMA) regressions using smatr v. 3.4.8 [48] to the same log-transformed data. When SMA fits showed extreme outliers, we used Huber's M estimation in place of least squares to decrease sensitivity to outliers [48]. Finally, we compared our cornea-body size allometry results to those from a recently published dataset including eight anuran families [28] (see Supplemental Methods in the electronic supplementary material).

### (f) Testing for ecological correlates of anuran eye size
We used residuals from the PGLS fit for log-transformed ED versus RM as a measure of phylogenetically corrected relative eye investment, with positive residuals indicating larger than average eyes for a given body mass and negative residuals indicating smaller than average eyes; results with eye investment relative to SVL are included in the electronic supplementary material. We quantified relative corneal investment using residuals from the PGLS fit for log-transformed CD versus ED. To make our plots

royalsocietypublishing.org/journal/rspb　　*Proc. R. Soc. B* **287**: 20201393

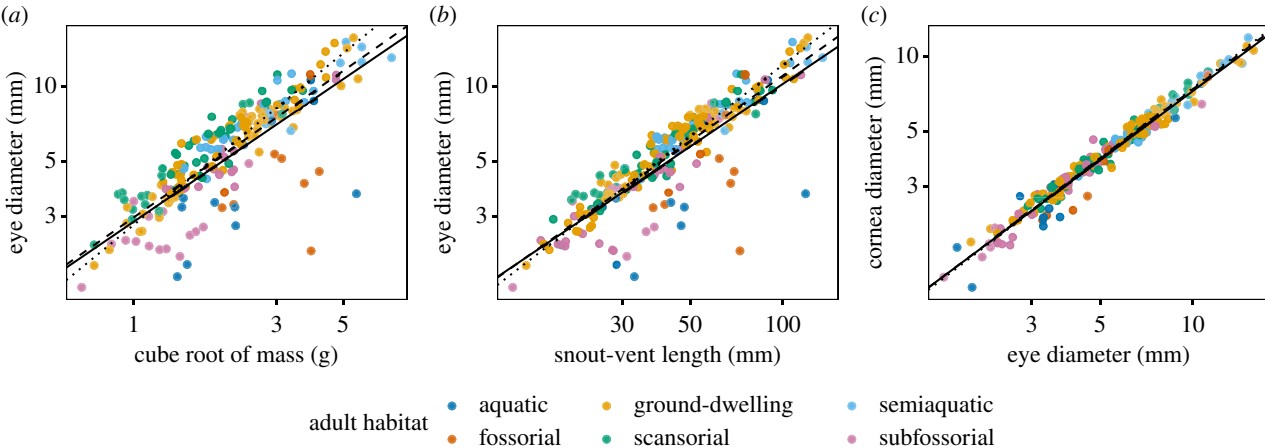

**Figure 2.** Allometric scaling of anuran eyes across 220 species. Transverse eye diameter scales hypoallometrically with both (*a*) the cube root of mass and (*b*) snout–vent length, exhibiting extreme negative outliers for species with small relative eye sizes. (*c*) Cornea diameter scales slightly hypoallometrically with eye diameter. Species means are coloured by adult habitat; lines show fits for PGLS (solid), SMA (dashed) and OLS (dotted) regressions. (Online version in colour.)

more intuitive, we exponentiated the PGLS residuals to show investment as a factor of the prediction for an animal's body size. For example, if a species had a phylogenetic residual of 0.2 in the PGLS of log-transformed ED versus RM, then its eye is approximately 1.6× (or $10^{0.2}$×) the diameter of an average anuran with the same mass.

We first used nonparametric Kruskal–Wallis tests to test whether EDs, relative eye investments or relative corneal investments differed among states for each of the six ecological traits we quantified (electronic supplementary material, table S1). Then, we tested whether ecology and relative eye size were correlated using phylogenetic ANCOVAs implemented through PGLS models in *caper* to test for the effect of each categorical ecological variable on eye size (ED), with body size (RM or SVL) as a covariate (e.g. ED ~ RM * adult habitat). We ran separate models for each ecological variable to avoid overparameterizing models and introducing rare categorical predictor combinations [44]. Finally, we used similar phylogenetic ANCOVA models to test for the effect of each ecological variable on corneal size relative to eye size (e.g. CD ~ ED * adult habitat). We plotted eye size, eye investment, and ecological states onto the phylogeny using *ape* v. 5.3 and *phytools* v. 0.6.99 in R [49,50].

## (g) Eye–body allometry in anurans compared to other vertebrates

We compiled published specimen data (*n* = 1210) for birds (Aves [15,51]), mammals (Mammalia [51,52]), ray-finned fishes (Actinopterygii [51]; Teleostei [20]; Acanthomorpha [53]), sharks and rays (Elasmobranchii [22,51]), and squamate reptiles (Squamata [51,54]; Colubridae [21]; Gekkonomorpha [19]) that used combinations of ED, AL, SVL, and body mass in scaling comparisons. We could not measure AL directly in museum specimens, so we used the highly correlated regression ($R^2$ = 0.96, *p* < 0.0001) of external ED versus dissected AL in fresh anuran specimens (*n* = 52; electronic supplementary material, figure S2A) to transform ED to AL in our museum specimen dataset.

We gathered species-level phylogenies matching published vertebrate data for birds [55], mammals [56,57], sharks and rays [58], and squamates [59] by downloading 1000 trees for each clade from VertLife [60] and then generating majority-rule consensus trees in Geneious Prime v.2019.2.3. We obtained a species-level phylogeny for ray-finned fishes from TimeTree [61], and used genus-level substitutions for 98/346 species in the dataset that were not present in the tree. For each clade and each scaling comparison, we pruned trees to match data for that comparison and

then fit PGLS regressions, as well as OLS regressions for comparison, to each clade using the same methods as for anurans (above).

## (h) Reproducibility

We conducted analyses in R v.3.6.1 [47] and provide annotated scripts to reproduce all analyses and figures on GitHub: https://github.com/knthomas/anuran-eye-size. Raw morphological data with specimen catalogue numbers, ecological trait data with supporting references, and compiled vertebrate eye and body size data are deposited on Dryad [62].

# 3. Results

## (a) Museum specimen measurements are reliable for eye–body allometry

In fresh anuran specimens, eye diameter measurements from whole dissected eyes were similar to, and highly correlated with, eye diameter measured externally prior to dissection in the same specimens (OLS: $R^2$ = 0.96, *n* = 55, s.e.res. = 0.04, $F_{1,53}$ = 1387, *p* < 0.0001, slope = 1.01, intercept = −0.02), indicating that external measurement of eye size is reasonable when specimens cannot be sampled destructively. Additionally, morphological data from preserved and fresh specimens yielded similar allometric relationships for eye–body scaling, suggesting preservation shrinkage is of minor concern in our museum specimen measurements (electronic supplementary material, figure S2). Finally, our measurements from museum specimens produced a similar allometric relationship between corneal diameter and body size to a published dataset measured photographically from fresh specimens (electronic supplementary material, figure S10) [28]. See Supplemental Results in the electronic supplementary material for details.

## (b) Anuran eyes scale hypoallometrically with body size

There was hypoallometric (slope < 1, negative allometry) interspecific scaling between anuran ED and RM (PGLS: slope ± s.e. = 0.82 ± 0.03; *t* = 29.4; d.f. = 1, 213; $R^2$ = 0.80; *p* < 0.0001; λ = 0.96) and ED and SVL (PGLS: slope ± s.e. = 0.84 ± 0.02; *t* = 34.3; d.f. = 1, 218; $R^2$ = 0.84; *p* < 0.0001; λ = 0.96; figure 2, electronic supplementary material, table S2). Similarly, CD scaled hypoallometrically with both RM (PGLS: slope ± s.e. =

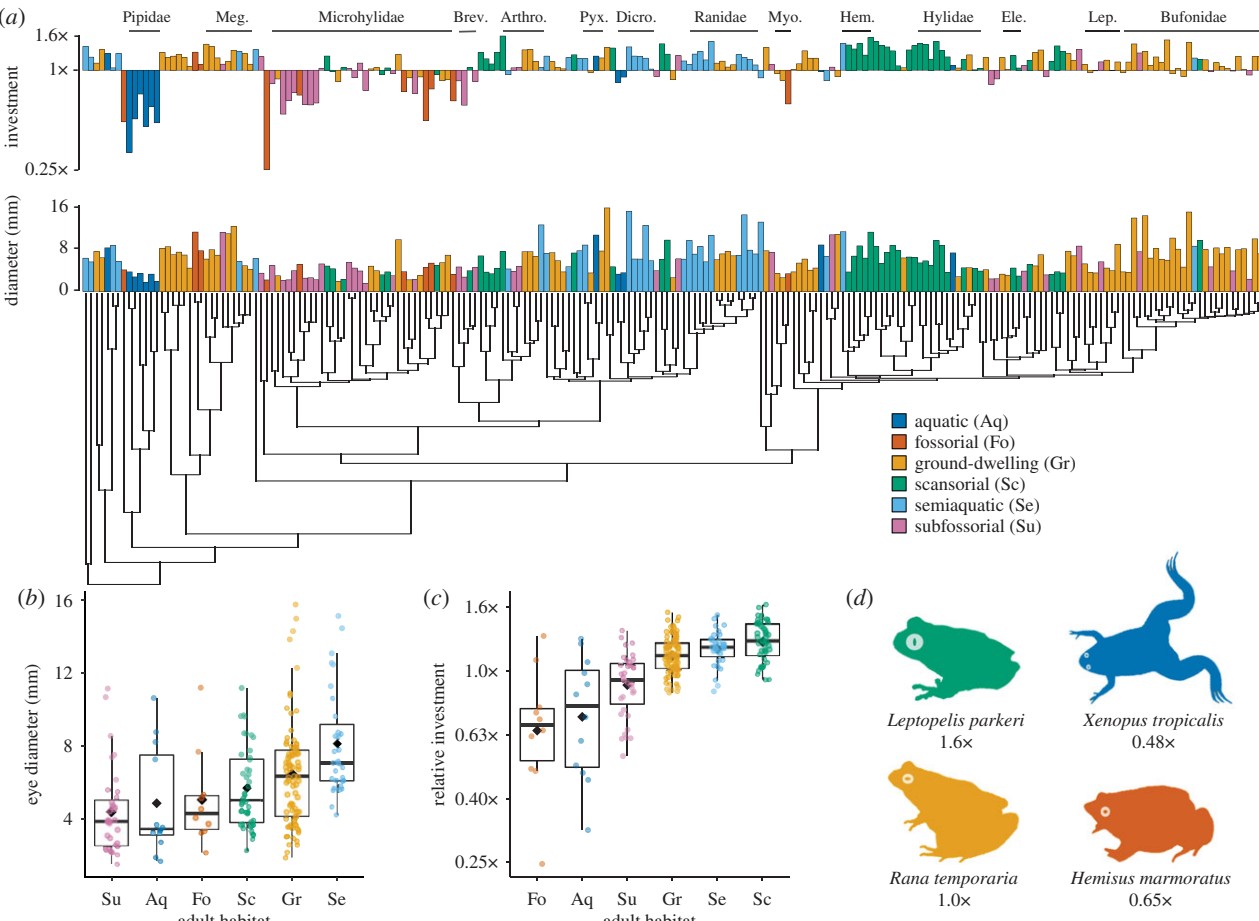

**Figure 3.** Eye size and investment vary significantly across adult habitats in anurans. (*a*) Phylogeny adapted from [36] shows species means for absolute eye diameter (ED) and relative eye investment. Eye investment represents the species residuals from the PGLS fit for log-transformed ED versus the cube root of body mass (RM), exponentiated so that species that fall precisely along the allometric fit have an eye investment of 1× the value predicted by fit, those that fall below have eye investments of less than 1× (small relative eye sizes), and those that fall above the fit have eye investments of greater than 1× (large relative eye sizes). Bars are coloured by adult habitat. Meg. = Megophryidae, Brev. = Brevicepitidae, Arthro. = Arthroleptidae, Pyx. = Pyxicephalidae, Dicro. = Dicroglossidae, Myo. = Myobatrachidae, Hem. = Hemiphractidae, Ele. = Eleutherodactylidae, Lep. = Leptodactylidae. (*b*) Absolute EDs and (*c*) relative eye investments differ significantly across adult habitats. Black diamonds indicate the mean and black bars the median for each state. (*d*) Silhouettes of four species (coloured by adult habitat) that exhibit differences in eye investment relative to body mass. (Online version in colour.)

0.74 ± 0.03; $t = 24.0$; d.f. = 1, 211; $R^2 = 0.73$; $p < 0.0001$; $\lambda = 0.88$) and SVL (PGLS: slope ± s.e. = 0.76 ± 0.03; $t = 28.1$; d.f. = 1, 216; $R^2 = 0.78$; $p < 0.0001$; $\lambda = 0.87$). CD was slightly hypoallometric with ED (PGLS: slope ± s.e. = 0.92 ± 0.01; $t = 72.6$; d.f. = 1, 216; $R^2 = 0.96$; $p < 0.0001$; $\lambda = 0.40$). Our two measures of body size (SVL versus RM) were highly correlated and nearly isometric (PGLS: slope ± s.e. = 0.98 ± 0.01; $t = 68.2$; d.f. = 1, 213; $R^2 = 0.96$; $p < 0.0001$; $\lambda = 0.75$). OLS and SMA models yielded similar fits to PGLS models, though slopes for SMA models were consistently the highest and PGLS the lowest (figure 2; electronic supplementary material, figure S1 and table S4). SMA tests indicated that the scaling of ED with RM, ED with SVL, and SVL with RM were isometric (electronic supplementary material, table S4).

## (c) Anuran eye size and investment correlate with ecological traits

We found that anuran eye size (both absolute and relative) differs significantly across species with different ecological traits (electronic supplementary material, table S7), and that eye size is correlated with ecology across the anuran phylogeny (electronic supplementary material, tables S8 and S9).

Species occupying different adult habitats have significantly different mean EDs (Kruskal–Wallis: $\chi^2 = 36.3$, d.f. = 5, $p < 0.0001$) and mean relative eye investments (Kruskal–Wallis: $\chi^2 = 68.3$, d.f. = 5, $p < 0.0001$; figure 3). Fossorial (5.1 mm), subfossorial (4.3 mm) and aquatic (4.9 mm) species had the smallest mean eye sizes as well as the smallest relative eye investments (fossorial = 0.65×, subfossorial = 0.91×, aquatic = 0.72×; figure 3). Scansorial anurans showed the highest investment (1.24×), followed by semiaquatic (1.18×) and ground-dwelling (1.12×) anurans. Finally, in a phylogenetic ANCOVA, both habitat ($F_{5,203} = 12.38$, $p < 0.0001$) and the interaction between habitat and RM ($F_{5,203} = 2.30$, $p = 0.046$) had significant effects on ED.

Species with different activity patterns showed significant differences in mean ED (Kruskal–Wallis: $\chi^2 = 6.37$, d.f. = 2, $p = 0.04$), but not relative eye investments (Kruskal–Wallis: $\chi^2 = 2.07$, d.f. = 2, $p < 0.36$; figure 4) or corneal investments (Kruskal–Wallis: $\chi^2 = 1.78$, d.f. = 2, $p < 0.41$; electronic supplementary material, figure S7). However, phylogenetic ANCOVA models indicated a significant effect of activity pattern on ED ($F_{2,173} = 49.6$, $p < 0.0001$; electronic supplementary material, table S8) and of activity pattern on CD ($F_{2,172} = 96.7$, $p < 0.0001$; electronic supplementary material, table S10).

Proc. R. Soc. B 287: 20201393

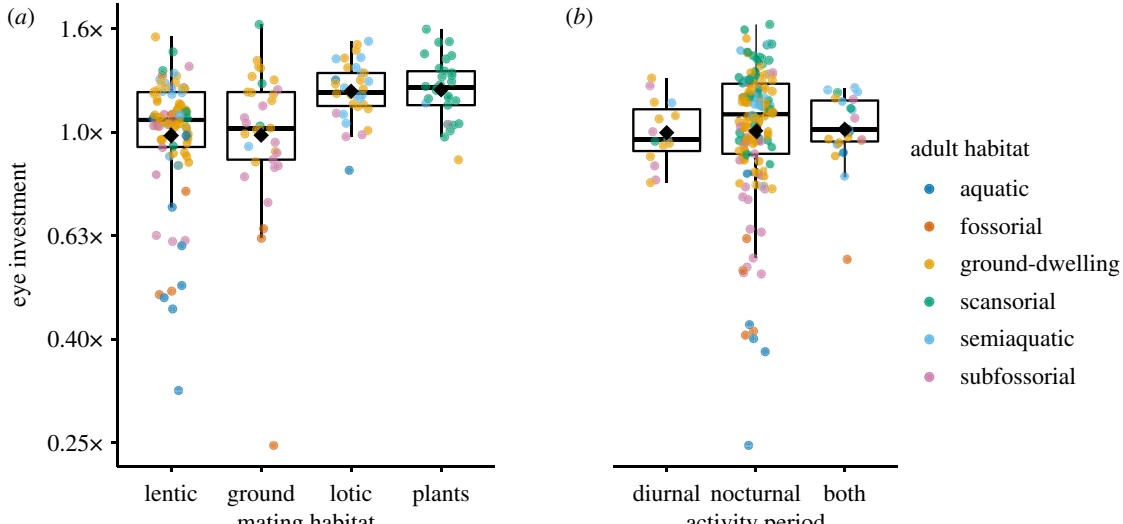

**Figure 4.** Eye investment differs significantly in phylogenetic ANCOVAs among anurans (*a*) utilizing different mating habitats and (*b*) active during different periods. Points are coloured by adult habitat, which showed the largest differences among states. Black diamonds indicate the mean and black bars the median for each state. (Online version in colour.)

Species mating in different habitats had significantly different mean EDs (Kruskal–Wallis: $\chi^2 = 24.6$, d.f. = 3, $p < 0.0001$) and eye investments (Kruskal–Wallis: $\chi^2 = 28.6$, d.f. = 3, $p = 0.002$), as well as different corneal investments (Kruskal–Wallis: $\chi^2 = 15.1$, d.f. = 3, $p < 0.0001$; electronic supplementary material, figure S5). Eye investment was positive in anurans mating on vegetation (1.21×) and in association with lotic water (1.20×), but negative in species mating in association with lentic water (0.99×) and on the ground (0.99×, figure 4). Phylogenetic ANCOVA models showed a significant effect of mating habitat on ED ($F_{3,170} = 3.71$, $p = 0.01$), as well as mating habitat on CD ($F_{3,170} = 5.15$, $p = 0.002$).

Life-history strategy and larval habitat both showed significant differences in mean ED and corneal investment across states, but not in relative eye investment (electronic supplementary material, figures S8 and S9 and table S7). However, phylogenetic ANCOVAs did not show significant effects of life-history strategy or larval habitat on eye size or corneal size (electronic supplementary material, tables S8 and S10). Sexual dichromatism did not show significant effects on eye or corneal size in any of our statistical tests (electronic supplementary material, tables S7–S10).

### (d) Anurans have relatively large eyes compared to other vertebrates

Comparisons across published measurements of vertebrate eye and body sizes consistently showed that anurans have large relative eye sizes and steep allometric slopes for the scaling of eye dimensions with body size (figure 5; electronic supplementary material, table S5). Anurans have large relative ALs among vertebrates, comparable to birds and some fishes, and a steeper allometric slope (0.73) for AL versus RM than birds (0.45), elasmobranchs (0.43), ray-finned fishes (0.69), and mammals (0.58). Anurans likewise exhibit larger relative ALs and a steeper allometric slope (0.85) for AL versus SVL than lizards (0.51). Teleost fishes exhibit higher variation in relative EDs than anurans, but the scaling of ED with RM in both clades exhibits a similar pattern of

repeated negative outliers representing species with small relative eye sizes. Finally, anurans (0.84), and squamates (geckos and colubrid snakes, 0.86) have similar slopes for the allometry of ED versus SVL, but anurans have much larger relative eye sizes due to body elongation in geckos and, especially, snakes.

## 4. Discussion

### (a) Ecological correlates of anuran eye size

Our results suggest that ecology has influenced the evolution of eye size in anurans. We found high variation in absolute eye size across anuran species and that adult habitat, activity period, mating habitat, life-history strategy and larval habitat showed significant differences in absolute eye size across states. Because eye size is correlated with body size (figure 2), differences in absolute eye size among ecotypes could result from selection on body size and/or relative eye investment. Absolute eye size is nevertheless important for visual ecology because it determines the optical limits of an eye, and means that anurans inhabiting different ecological niches likely have different optical constraints due to eye size [2,52]. We also found evidence of high variation in relative eye investment across Anura. Mean eye investment differed significantly among anurans utilizing different adult habitats and mating habitats. Further, both of these traits and activity pattern were correlated with eye size when correcting for body size and phylogeny, suggesting a role for ecology in eye size evolution.

The largest differences in eye investments among ecological states were found for adult habitats, where, for a given body mass, mean eye investments ranged from 1.24× (in scansorial anurans) to 0.65× (in fossorial anurans) the eye size predicted by the PGLS fit (figure 3). Large eyes may benefit scansorial anurans by accommodating fast temporal resolutions during jumping (as in flying birds [63]), high acuity in visually complex arboreal habitats (as in reef fishes [64]), and/or colour discrimination in low-light conditions [65,66]. Fossorial anurans probably have reduced eye investments as adaptations to dark and/or abrasive

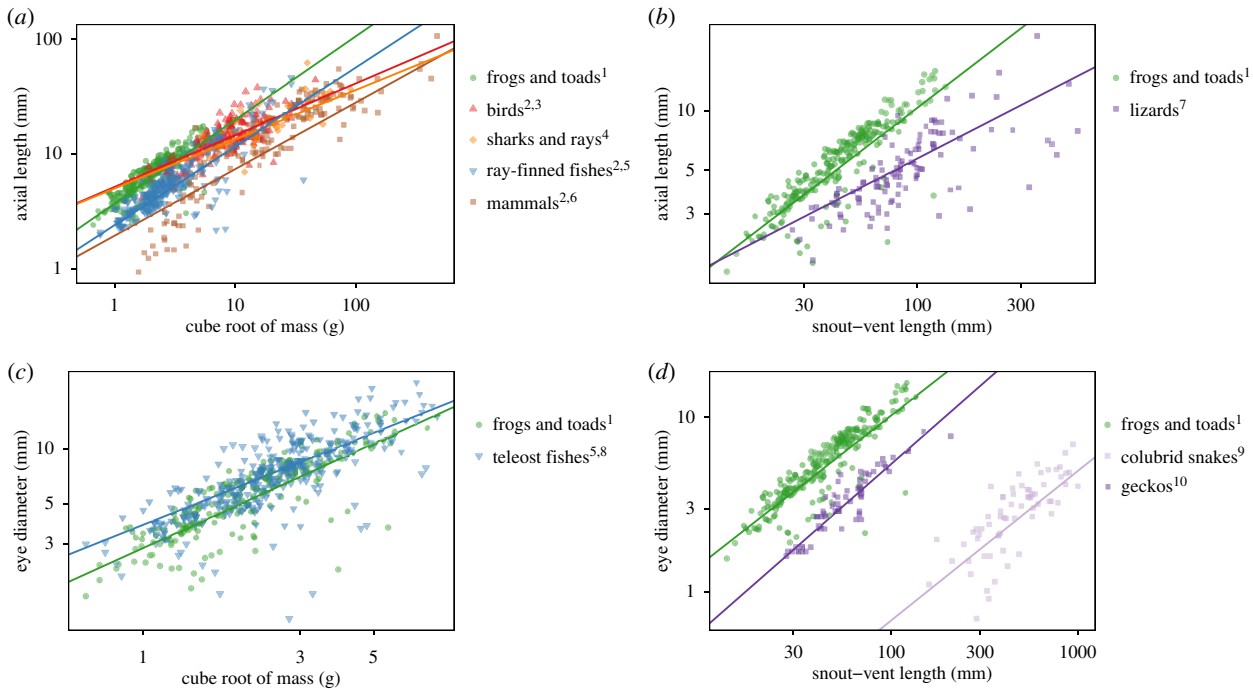

**Figure 5.** Anurans show large relative eye sizes and steep allometric slopes for eye–body scaling among vertebrate groups with published data for (*a*) axial length (AL) versus the cube root of mass (RM), (*b*) AL versus snout–vent length (SVL), (*c*) eye diameter (ED) versus RM, and (*d*) ED versus SVL. Data from [1]this study, [2][51], [3][15], [4][22], [5][20], [6][52], [7][54], [8][53], [9][21], [10][19]. (Online version in colour.)

habitats. Substantial reductions in eye size also occur in fossorial mammals (e.g. rodents, moles, mole rats [67]), fishes (e.g. eels [68]), caecilian amphibians [69] and reptiles (e.g. lizards [70]). Interestingly, mean eye investment increased from fully fossorial (0.65×) to subfossorial (0.91×) to non-fossorial (ground-dwelling, semiaquatic, scansorial) anurans (1.12× – 1.24×), strongly suggesting that degree of fossoriality is an important determinant of relative eye size.

We also found large reductions in relative eye size in aquatic habitats. Fully aquatic anurans had a low mean eye investment (0.72×), while semiaquatic anurans did not (1.18×). Although animals inhabiting clear water often have high eye investments to compensate for the quicker attenuation of light in water than air (e.g. pelagic fishes [64]), the aquatic anurans we sampled often occur in turbid environments, hunt and breed below the water surface, and sometimes bury in mud or litter at the bottom of ponds and pools [71]. These conditions may favour alternate sensory modalities, such as the adult lateral line systems present in several aquatic groups (figure 1*a* [*Xenopus*] and 1*f* [*Lepidobatrachus*]) [72,73]. By contrast, semiaquatic anurans are active on land, and many sit in the water with their eyes above the surface so that vision is occurring in air [71]. This likely explains their similarity to ground-dwelling frogs rather than aquatic frogs in terms of eye investment.

We found significant effects of activity pattern on relative eye size and relative corneal size among anurans, but these differences were small in magnitude compared to differences across adult habitats. Many nocturnal vertebrates show increased relative eye sizes or cornea diameters to maximize visual sensitivity (e.g. birds [16,74]; predatory primates [17]; reef fish [20]), though sometimes this is significant only when activity pattern is combined with habitat to assign overall photopic versus scotopic lifestyles (e.g. lizards [54]). However, non-primate mammals show predominantly nocturnal eye

morphologies despite the repeated evolution of diurnality, which is thought to represent a 'nocturnal bottleneck' resulting from a long evolutionary history of nocturnality and limited selection pressure to reduce relative corneal size when a dynamic pupil can achieve the same function [18,75]. Because the common anuran ancestor was likely nocturnal and the majority of extant anurans retain this activity period [76], relative eye sizes and corneal proportions in anurans may also represent a nocturnal bottleneck.

Vision may be important in anuran breeding ecology, evidenced by significant differences in eye investment across mating habitats. Anurans mating in vegetation or in association with lotic water had positive mean eye investment, while those mating on the ground or in association with lentic water had negative investment. Large eyes may be beneficial in mating for conspecific recognition and visual signalling, and these results are consistent with evidence that anurans breeding in or near lotic water are more likely to employ visual signalling because the noise of rushing water inhibits auditory signal transmission [26], though this merits further study. However, sexual dichromatism is thought to function in sexual signalling among some anurans [31,32]. Yet, we did not detect significant effects of sexual dichromatism on anuran eye size. Further work on the spectral sensitivities of frogs relative to their body colorations and the light environments they mate in will improve our understanding of potential visual signals in anuran breeding ecology and how this relates to relative eye size.

## (b) Anuran eyes compared to other vertebrates

Our study revealed that anurans have large relative eye sizes compared to other vertebrates (figure 5). For a given body mass, anurans had similar axial lengths to birds (figure 5*a*), which previously have shown the highest investments in eye

size among extant vertebrates [51]. Anurans also show scaling of transverse eye diameter and mass that, similar to teleost fishes, shows repeated, dramatic reductions in eye diameter; a pattern not observed in available amniote datasets (birds, mammals, squamates). Relative eye size is often used as an indirect measure of the importance of vision to a taxon (e.g. [29,64,77]); thus our results suggest that vision is highly important in anurans, yet anuran visual ecology is relatively understudied compared with other vertebrate groups. Further work on the sensitivity, acuity and temporal resolution of diverse amphibians is needed to better understand the functions of large eyes in Anura.

Generally, vertebrate eyes are thought to show hypoallometric (slope < 1) interspecific scaling with body size, so that smaller species have relatively bigger eyes than larger species [51,78,79]. We found that anuran eyes are consistent with this pattern, but exhibit high slopes and intercepts compared to other groups, so that relatively large eyes occur across all body sizes. Previous studies on geckos [19] and snakes [21] found similarly steep slopes for the scaling of eye diameter with snout–vent length, though the elongation of squamates (and snakes in particular) result in smaller eye sizes relative to body sizes in these taxa.

## (c) Consistency with prior work and future directions

Studies of eye size across vertebrates have used data from species descriptions, scaled photographs, fresh specimens and/or preserved specimens, and this may introduce error when comparing across studies and groups. Here, we derived similar eye scaling relationships from a small dataset we collected from fresh specimens ($n = 67$) and a larger dataset we collected from museum specimens ($n = 640$). Additionally, our museum data yielded similar results for cornea-body size allometry to fits derived from an independent dataset that used photographic measurements of corneal diameter [28]. This suggests that, at least in anurans, error introduced by different measurement techniques and specimen treatments is not great enough to mask underlying allometric patterns. Our novel finding that relative eye investment correlates with adult habitat, mating habitat and activity pattern was different from previously reported results [28], likely due to increased sampling and the inclusion of broader ecological diversity (e.g. fossorial species, diurnal species) in our analyses.

Although the evolutionary allometry of eyes among vertebrates is well studied, little is known about the ontogenetic allometry of eyes within species that leads to observed interspecific patterns among adults. In biphasic anurans this is especially intriguing, as morphological evolution can be decoupled for larvae and adults [80]. We found no significant differences in adult eye investment among species with different life-history strategies or that occupy different habitats as larvae. However, because anurans exhibit a rich diversity of growth strategies and larval ecologies, examining how ontogenetic eye–body allometry produces observed interspecific differences in adult eye sizes may be particularly fruitful in this group.

This study is an important first step in understanding the evolution of eye size in frogs and toads, because we have quantified variation in relative eye size among anurans and found ecological correlates of eye size across the anuran phylogeny. However, ecological correlates do not necessarily drive the evolution of a trait. For example, although geckos show correlations between activity period and relative eye size [19], modelling of transitions in adaptive regimes for eye size across the gecko phylogeny indicate that activity period alone does not explain regime shifts [81]. We hope that our data and findings will spur exploration into the adaptive evolutionary forces driving changes in anuran eye size.

## 5. Conclusion

Our taxonomically broad examination of anuran amphibian eye size, eye investment and corneal investment across species demonstrates, for the first time, that ecology is correlated with visual morphology in this group. Notably, adult habitat was associated with both increased investment (complex habitats; scansorial habits) and extreme divestment (scotopic environments; fossorial and aquatic habits) in eye size among anurans. Relative eye and corneal sizes were correlated with activity pattern but with small magnitude differences, indicating a potential nocturnal bottleneck in the evolution of anuran eyes. Additionally, we found that anurans have some of the largest eyes relative to their body size among sampled vertebrates, highlighting the importance of considering diverse vertebrate lineages to gain a comprehensive understanding of visual system evolution in vertebrates.

Ethics. Live anurans were obtained through fieldwork in French Guiana (permit RAA: R03-2018-06-12-006) and via the pet trade (NHM licence NE: WML-OR04). Institutions and catalogue numbers for all museum specimens used in this study are included in the data files deposited on Dryad.

Data accessibility. The datasets supporting this article are available from the Dryad Digital Repository: https://dx.doi.org/10.5061/dryad. 1zcrjdfq7 [62], and code to replicate all analyses and figures are available on GitHub (https://github.com/knthomas/anuran-eye-size).

Authors' contributions. K.N.T., J.W.S., R.C.B., M.K.F. and D.J.G. conceived the project. K.N.T. collected morphological data from museum specimens selected and taxonomically confirmed by J.W.S.; D.J.G. measured four additional specimens at the Bombay Natural History Society (Mumbai, India). J.W.S. and D.J.G. led field expeditions and, with K.N.T., collected morphological data from fresh specimens. K.N.T. implemented the analyses and, with J.W.S., drafted the manuscript. All authors participated in regular discussion of the project and contributed to ecological trait assignments and revision of the manuscript.

Competing interests. We declare we have no competing interests.

Funding. This work was supported by grants from the Natural Environment Research Council, UK (grant no. NE/R002150/1) and the National Science Foundation, USA (grant no. DEB #1655751).

Acknowledgements. We thank Bryan Stuart for hosting K.N.T. at the NCMNS and Rahul Khot for hosting D.J.G. at BNHS for specimen measurements, Centre National de la Recherche Scientifique (Cayenne, French Guiana) for facilitating fieldwork for J.W.S. and D.J.G., and Gabriela Bittencourt-Silva, Mark Wilkinson, Kim Roelants, Sunita Janssenwillen, Hannah Augustÿnen, Simon Maddock, Christian Cox, and Jon and Krittee Gower for collecting fresh specimens. Stephen Mahony, Simon Loader, Mark-Oliver Rödel, Rafael de Sá, Fred Kraus, Jim Labisko, Deepak Veerappan, Bryan Stuart, Yodchaiy Chuaynkern, Alan Channing, and the NHM Herpetology Group provided guidance in assigning species to ecological categories. Natalie Cooper provided valuable assistance with comparative methods in R and insightful comments on the manuscript, with three anonymous reviewers. We are deeply grateful to the long history of collectors and curators who have built and safeguarded natural history collections, making this and countless other research possible.

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
