## [Reviewer comments · Proceedings of the Royal Society B: Biological Sciences]

Review History

RSPB-2019-2420.R0 (Original submission)

Review form: Reviewer 1

Recommendation

Reject – article is scientifically unsound

Scientific importance: Is the manuscript an original and important contribution to its field?

Marginal

General interest: Is the paper of sufficient general interest?

Good

Quality of the paper: Is the overall quality of the paper suitable?

Poor

Is the length of the paper justified?

Yes

Should the paper be seen by a specialist statistical reviewer?

No

Do you have any concerns about statistical analyses in this paper? If so, please specify them explicitly in your report.

Yes

It is a condition of publication that authors make their supporting data, code and materials available - either as supplementary material or hosted in an external repository. Please rate, if applicable, the supporting data on the following criteria.

Is it accessible?

Yes

Is it clear?

Yes

Is it adequate?

Yes

Do you have any ethical concerns with this paper?

No

Comments to the Author

I have read the manuscript on eye size in frogs with great interest. The visual system of frogs is indeed not well known, and any new data promise exciting findings. However, there are several shortcomings in the design of the study that question the validity of the results.

A major concern is the false equivalence of considering both traditional statistical methods and phylogenetically informed approaches. It has been demonstrated many times that one must account for phylogenetic covariance in the data when dealing with interspecific datasets. Why perform traditional analyses when a time-calibrated phylogeny is available? All SMA analyses and results should be considered invalid (both for frogs and across vertebrates). The potentially interesting finding of an isometric slope of eye growth compared to body size, which could have been a major result of the paper, is not supported: the possible isometry is advertised in the abstract, but the phylogenetically informed results demonstrate that isometry is rejected.

In addition, the choice of phylogenetic comparative methods is inadequate. There are many modern comparative approaches that allow tests for adaptive evolution in a modeling framework, routinely employed in many papers on trait evolution (OUwie, mvMORPH, I1ou, bayou, and many other R packages). A thorough analysis of eye evolution in frogs should go far beyond a PGLS (which would need revision to be a true phylogenetic ANCOVA).

There are additional concerns regarding the data collection and study design. To enable large-scale comparisons across different clades some eye diameter data were converted into length data which I would think is problematic because not all eyes have the same ellipticity. Also, there is one comparison where snakes, geckos, and frogs are compared by putting SVL length on the X-axis. That comparison does not make much sense because snakes are so elongated, and even geckos are much more elongated than frogs are.

Finally, the questions of the manuscripts are not guided by clear hypotheses. What are the exact implications of eye size on vision and what eye size changes are expected because of habitats? An obvious pathway may be a test of MacIver's 2017 buena vista hypothesis (eye size changes between water and air/land).

Review form: Reviewer 2

Recommendation

Accept with minor revision (please list in comments)

Scientific importance: Is the manuscript an original and important contribution to its field?
Good

General interest: Is the paper of sufficient general interest?
Good

Quality of the paper: Is the overall quality of the paper suitable?
Good

Is the length of the paper justified?
Yes

Should the paper be seen by a specialist statistical reviewer?
No

Do you have any concerns about statistical analyses in this paper? If so, please specify them explicitly in your report.
Yes

It is a condition of publication that authors make their supporting data, code and materials available - either as supplementary material or hosted in an external repository. Please rate, if applicable, the supporting data on the following criteria.

Is it accessible?
Yes

Is it clear?
Yes

Is it adequate?
Yes

Do you have any ethical concerns with this paper?
No

Comments to the Author

This manuscript, entitled "Eye size and investment in frogs and toads correlate with adult habitat and breeding ecology", aims to unravel the correlation between relative eye size and different ecological and behavioural traits. The authors collected several morphological measurements and ecological and behavioural categorical traits from 220 species comprising 55 anuran families. They also compare anuran relative eye sizes with other vertebrate groups. They found that relative eye size is significantly correlated (without phylogenetic correction) with habitat, and that eye size is proportionally inverse to fossoriality degree. They also found a significant correlation between relative eye size and mating habitat and sexual dichromatism. Finally, the authors found that frogs have relatively large eyes compared to other vertebrates.

Overall, I quite enjoyed reading this manuscript. It is well written, the authors have performed a lot of sound analyses, and it tackles a very important and exciting question on morphological adaptation. Therefore, I think this will be a very interesting contribution and I recommend it to be published in Proceedings B.

However, I have a few suggestions, particularly around the use of phylogenetic corrections in the analyses (and subsequent discussion of the results). Incorporating the phylogeny should be a selling point of this paper, since previous studies did not do it. However, the authors mostly focused on the non-phylogenetically corrected analyses (and results interpretation). As well as

that, the authors should make it clearer that fish have similar relative eye size ‘trend’ as fish. In Table S6 the authors present SM fits for eye scaling in different vertebrate groups, and the slope 95% C.I. of anurans and fish overlap. I think they should develop this in the discussion and change ‘vertebrate’ for ‘tetrapod’. I have provided a few comments that I hope will help to improve this paper. However, some of the comments I am giving are a personal preference, so while I hope they will help to improve the paper, I do not expect the authors to perform or reply to all of them.

ABSTRACT:

19 – the authors should clarify whether they are referring to non-phylogenetically corrected results or not. It is unclear here.

27 – Most of the results focus on non-phylogenetically corrected data, so it would not be suitable to infer and interpret macroevolutionary patterns.

INTRODUCTION:

36 - Reference needed

41-49 – This paragraph provides important technical background information, but it isn’t explained as well as it could be. The authors might want to consider re-writing it.

43 – Reference needed

45 – Is this a comparison to a camera? I think it needs a little bit more clarification

79 – It might be good re-phrasing this sentence in a more positive way, Also, could the authors add other differences to that paper? I think this manuscript will be a very important new contribution, but here it sounds a bit like the main difference between their manuscript and the paper they cite is ‘adding extra sampling’.

METHODS:

87 - The authors should define RM, AL and AD in this section.

100 – It might be better to talk about sampling sizes, etc. earlier in the M&M section.

118 - Please refer to a specific table.

122 – It would be useful for the readers if the authors explain why they used SMA instead of ordinary least squares (OLS). It might be good to mention degree of asymmetry, bias in the slope estimates of SMA vs. OLS, etc.

154 – The authors need to define AL. I was not able to find it anywhere in the methods.

156 - Define AD.

174 – Provide more details.

DISCUSSION

310-313 – See my main comment regarding phylogenetic correction. The phylogenetically-corrected result is rushed in this paragraph. I think it should be discussed further. As well as that, the last sentence sounds a bit like they are not doing it because previous studies did not use phylogenetic correction.

318 – which states?

319 “with” body size instead of “to”

328 – “where given body mass” – strange wording

337 – And scansorial?

356-358 – This needs clarification

395 – I think it is important to further explain this bit (“... difference in how we categorized states”)

FIGURES, TABLES, AND SUPPLEMENTARY MATERIAL

The authors might want to consider changing the colour scheme used. I struggled to see each of the categories in some of the plots.

Fig. 2 – Be consistent (use RM instead of body mass. Similarly ED instead of eye size.

Fig. 4, Fig. S4-S9 – Since the taxonomic sampling presented here is uneven across the anuran

(complete phylogeny), this will greatly affect the ancestral state reconstruction. I think the authors should point out this bias.

Decision letter (RSPB-2019-2420.R0)

02-Dec-2019

Dear Dr Thomas:

I am writing to inform you that we have now obtained responses from referees on manuscript RSPB-2019-2420 entitled "Eye size and investment in frogs and toads correlate with adult habitat and breeding ecology" which you submitted to Proceedings B.

Unfortunately, on the advice of the Associate Editor and the referees, your manuscript has been rejected following full peer review. Competition for space in Proceedings B is currently extremely severe, as many more manuscripts are submitted to us than we have space to print. We are therefore only able to publish those that are exceptional, convincing and present significant advances of broad interest, and must reject many good manuscripts.

Please find below the comments received from referees concerning your manuscript, not including confidential reports to the Editor. I hope you may find these useful should you wish to submit your manuscript elsewhere.

We are sorry that your manuscript has had an unfavourable outcome, but would like to thank you for offering your work to Proceedings B.

Sincerely,
Dr Sasha Dall
mailto:proceedingsb@royalsociety.org

Associate Editor

Comments to Author:

This study looks at eye size and investment in anurans, compares scaling relationships with other vertebrate groups and examines the relationship between eye investment and habitat. A strength of this study is the sample size and validation of the methods (comparison of museum and fresh specimens, reproducibility). However, analyses and results comprise a mix of phylogenetically corrected and uncorrected analyses. This makes results confusing and difficult to interpret because robust inferences can only be drawn from phylogenetically corrected analyses. I realise that phylogenetically uncorrected analyses were done to enable comparison across vertebrate groups and to previous studies, but phylogenetic information should be available for these groups, and with no phylogenetic correction, the comparison doesn't mean much. The biological significance of comparing the relationship between eye size and SVL in anurans, snakes and lizards is also unclear because of the difference in body shape (elongation). Also, as I understand it, analyses relating eye investment to habitat were done using phylogenetically corrected relative eye size but tests for differences between habitats were not phylogenetically corrected. This means that phylogenetic non-independence in habitat was not accounted for, again making results difficult to interpret. Lastly, as reviewer 1 points out, the study would benefit from articulating clear hypotheses about relationships between eye investment and habitat. Because the biological significance of the results is unclear given the analyses, I cannot recommend acceptance in PRSLB. However, the reviewers have provided valuable feedback and I'm sure that this impressive dataset can be used to make a valuable contribution to our understanding of the evolution of eye investment in anurans.

Reviewer(s)' Comments to Author:

Referee: 1

Comments to the Author(s)

I have read the manuscript on eye size in frogs with great interest. The visual system of frogs is indeed not well known, and any new data promise exciting findings. However, there are several shortcomings in the design of the study that question the validity of the results.

A major concern is the false equivalence of considering both traditional statistical methods and phylogenetically informed approaches. It has been demonstrated many times that one must account for phylogenetic covariance in the data when dealing with interspecific datasets. Why perform traditional analyses when a time-calibrated phylogeny is available? All SMA analyses and results should be considered invalid (both for frogs and across vertebrates). The potentially interesting finding of an isometric slope of eye growth compared to body size, which could have been a major result of the paper, is not supported: the possible isometry is advertised in the abstract, but the phylogenetically informed results demonstrate that isometry is rejected.

In addition, the choice of phylogenetic comparative methods is inadequate. There are many modern comparative approaches that allow tests for adaptive evolution in a modeling framework, routinely employed in many papers on trait evolution (OUwie, mvMORPH, I1ou, bayou, and many other R packages). A thorough analysis of eye evolution in frogs should go far beyond a PGLS (which would need revision to be a true phylogenetic ANCOVA).

There are additional concerns regarding the data collection and study design. To enable large-scale comparisons across different clades some eye diameter data were converted into length data which I would think is problematic because not all eyes have the same ellipticity. Also, there is one comparison where snakes, geckos, and frogs are compared by putting SVL length on the X-axis. That comparison does not make much sense because snakes are so elongated, and even geckos are much more elongated than frogs are.

Finally, the questions of the manuscripts are not guided by clear hypotheses. What are the exact implications of eye size on vision and what eye size changes are expected because of habitats? An obvious pathway may be a test of MacIver's 2017 buena vista hypothesis (eye size changes between water and air/land).

Referee: 2

Comments to the Author(s)

This manuscript, entitled "Eye size and investment in frogs and toads correlate with adult habitat and breeding ecology", aims to unravel the correlation between relative eye size and different ecological and behavioural traits. The authors collected several morphological measurements and ecological and behavioural categorical traits from 220 species comprising 55 anuran families. They also compare anuran relative eye sizes with other vertebrate groups. They found that relative eye size is significantly correlated (without phylogenetic correction) with habitat, and that eye size is proportionally inverse to fossoriality degree. They also found a significant correlation between relative eye size and mating habitat and sexual dichromatism. Finally, the authors found that frogs have relatively large eyes compared to other vertebrates.

Overall, I quite enjoyed reading this manuscript. It is well written, the authors have performed a lot of sound analyses, and it tackles a very important and exciting question on morphological adaptation. Therefore, I think this will be a very interesting contribution and I recommend it to be published in Proceedings B.

However, I have a few suggestions, particularly around the use of phylogenetic corrections in the analyses (and subsequent discussion of the results). Incorporating the phylogeny should be a selling point of this paper, since previous studies did not do it. However, the authors mostly focused on the non-phylogenetically corrected analyses (and results interpretation). As well as that, the authors should make it clearer that fish have similar relative eye size 'trend' as fish. In

Table S6 the authors present SM fits for eye scaling in different vertebrate groups, and the slope 95% C.I. of anurans and fish overlap. I think they should develop this in the discussion and change 'vertebrate' for 'tetrapod'. I have provided a few comments that I hope will help to improve this paper. However, some of the comments I am giving are a personal preference, so while I hope they will help to improve the paper, I do not expect the authors to perform or reply to all of them.

ABSTRACT:

19 - the authors should clarify whether they are referring to non-phylogenetically corrected results or not. It is unclear here.

27 - Most of the results focus on non-phylogenetically corrected data, so it would not be suitable to infer and interpret macroevolutionary patterns.

INTRODUCTION:

36 - Reference needed

41-49 - This paragraph provides important technical background information, but it isn't explained as well as it could be. The authors might want to consider re-writing it.

43 - Reference needed

45 - Is this a comparison to a camera? I think it needs a little bit more clarification

79 - It might be good re-phrasing this sentence in a more positive way, Also, could the authors add other differences to that paper? I think this manuscript will be a very important new contribution, but here it sounds a bit like the main difference between their manuscript and the paper they cite is 'adding extra sampling'.

METHODS:

87 - The authors should define RM, AL and AD in this section.

100 - It might be better to talk about sampling sizes, etc. earlier in the M&M section.

118 - Please refer to a specific table.

122 - It would be useful for the readers if the authors explain why they used SMA instead of ordinary least squares (OLS). It might be good to mention degree of asymmetry, bias in the slope estimates of SMA vs. OLS, etc.

154 - The authors need to define AL. I was not able to find it anywhere in the methods.

156 - Define AD.

174 - Provide more details.

DISCUSSION

310-313 - See my main comment regarding phylogenetic correction. The phylogenetically-corrected result is rushed in this paragraph. I think it should be discussed further. As well as that, the last sentence sounds a bit like they are not doing it because previous studies did not use phylogenetic correction.

318 - which states?

319 "with" body size instead of "to"

328 - "where given body mass" - strange wording

337 - And scansorial?

356-358 - This needs clarification

395 - I think it is important to further explain this bit ("... difference in how we categorized states")

FIGURES, TABLES, AND SUPPLEMENTARY MATERIAL

The authors might want to consider changing the colour scheme used. I struggled to see each of the categories in some of the plots.

Fig. 2 - Be consistent (use RM instead of body mass. Similarly ED instead of eye size.

Fig. 4, Fig. S4-S9 - Since the taxonomic sampling presented here is uneven across the anuran (complete phylogeny), this will greatly affect the ancestral state reconstruction. I think the authors should point out this bias.

Author's Response to Decision Letter for (RSPB-2019-2420.R0)

See Appendix A.

RSPB-2020-1393.R0

Review form: Reviewer 1

Recommendation

Accept with minor revision (please list in comments)

Scientific importance: Is the manuscript an original and important contribution to its field?

Good

General interest: Is the paper of sufficient general interest?

Excellent

Quality of the paper: Is the overall quality of the paper suitable?

Good

Is the length of the paper justified?

Yes

Should the paper be seen by a specialist statistical reviewer?

No

Do you have any concerns about statistical analyses in this paper? If so, please specify them explicitly in your report.

Yes

It is a condition of publication that authors make their supporting data, code and materials available - either as supplementary material or hosted in an external repository. Please rate, if applicable, the supporting data on the following criteria.

Is it accessible?

Yes

Is it clear?

Yes

Is it adequate?

Yes

Do you have any ethical concerns with this paper?

No

Comments to the Author

I am very happy with the modified version of this manuscript. The analyses are very detailed and the way how scripts were made available is exemplary. I have a few suggestions that should be addressed:

- 1) PGLS is currently implemented with a Brownian Motion correlation structure. Given that the data suggest that adaptive processes may shape evolutionary pattern, an underlying OU correlation structure may be more appropriate.
- 2) Ancestral state reconstruction are currently implemented with a single (equal) rate. One should check whether this is supported by the data.
I also would like to see more detail on the specific effect of eye size on visual performance, specifically with respect to the sensitivity of an eye to extended light sources vs point light sources. A larger aperture in absolute terms will result in better light sensitivity, while absolute eye size does not influence sensitivity to extended light sources (all else being equal).
- 3) As for cornea size, in how far can it be demonstrated that cornea size is correlated with maximum pupil diameter? Corneal diameter itself does not have a direct effect on visual performance. Corneal curvature would, however.
- 4) Lastly, I was wondering whether Ritland's 1982 dissertation on eye size across vertebrates should be cited. To the best of my knowledge Ritland's work was never published but the dissertation is available and has been cited in other comparative eye studies.

Review form: Reviewer 3 (Emma Sherratt)

Recommendation

Accept with minor revision (please list in comments)

Scientific importance: Is the manuscript an original and important contribution to its field?

Excellent

General interest: Is the paper of sufficient general interest?

Excellent

Quality of the paper: Is the overall quality of the paper suitable?

Excellent

Is the length of the paper justified?

Yes

Should the paper be seen by a specialist statistical reviewer?

No

Do you have any concerns about statistical analyses in this paper? If so, please specify them explicitly in your report.

No

It is a condition of publication that authors make their supporting data, code and materials available - either as supplementary material or hosted in an external repository. Please rate, if applicable, the supporting data on the following criteria.

Is it accessible?

Yes

Is it clear?

Yes

Is it adequate?

Yes

Do you have any ethical concerns with this paper?

No

Comments to the Author

I have reviewed the manuscript entitled “Eye size and investment in frogs and toads correlate with adult habitat, activity pattern, and breeding ecology” and find it to be a very well written and interesting study. It is a very important contribution to the literature, addressing a prominent gap in our understanding of species-level diversity patterns in vertebrates. I think it is an excellent paper, polished and without any major issues. Statistically, it is sound and extensive. The figures are beautiful, clear and easy to follow. The dataset is very impressive and commendable. I have only minor comments suggested for clarity.

The terminology behind allometry and allometric scaling as viewed from regression analyses could have some consideration:

“Developmental allometry” – usually called ontogenetic allometry

Hypoaallometric – also known as negative allometry; could be good to clarify this at first use for readers used to that terminology

“high allometric slopes” – I think the authors are referring to the intercept differences here, and not the angle of the slope. Better to clarify, since it is ambiguous in places.

L66 (and 68): maybe change to say “eye size (absolute and relative to body size)” here, since “and allometric scaling” is ambiguous. Or make it clear that allometric scaling belongs to the eye size.

Methods: sampling – I don’t think that the sentences L105-108 needs to be a separate paragraph. This information can fit in after the museum collection sentences on L97 and into the measurements taken sentences after it.

L111 – “downloaded from dryad” unnecessary. Can simply have [31, 33] after phylogeny.

Fig. 3 – really beautiful figure. On the empty space, stylistic suggestion: could a schematic/silhouettes be put here showing eye sizes in two extreme frogs (ground dwelling vs scansorial)?

caption – please add in a brief explanation of what is mean of eye investment here. These two suggestions make the figure stand alone and would make an excellent graphical abstract.

I especially like the part of the predictions regarding eye size in adults where their larvae have a particular need for high visual acuity. I wonder if it is worth adding a sentence in the discussion clarifying that due to the adaptive decoupling hypothesis, eye size may not be a heritable trait across metamorphic boundary. Though to my knowledge of this literature, there are no papers that have looked at the heritability of eye size.

Finally, I’m glad to see the support for museum specimens highlighted here; the accuracy analysis of preserved and fresh material is not only an important finding, but also commendable and leading by example.

Decision letter (RSPB-2020-1393.R0)

03-Aug-2020

Dear Dr Thomas:

Your manuscript has now been peer reviewed and the reviews have been assessed by an Associate Editor. The reviewers’ comments (not including confidential comments to the Editor)

and the comments from the Associate Editor are included at the end of this email for your reference. As you will see, the reviewers and the Editors have raised some concerns with your manuscript and we would like to invite you to revise your manuscript to address them.

Research ethics:

Use of animals and field studies:

It is a condition of publication that you make available the data and research materials supporting the results in the article (<https://royalsociety.org/journals/authors/author-guidelines/#data>). Datasets should be deposited in an appropriate publicly available repository and details of the associated accession number, link or DOI to the datasets must be included in the Data Accessibility section of the article (<https://royalsociety.org/journals/ethics-policies/data-sharing-mining/>). Reference(s) to datasets should also be included in the reference list of the article with DOIs (where available).

Please submit a copy of your revised paper within three weeks. If we do not hear from you within this time your manuscript will be rejected. If you are unable to meet this deadline please let us know as soon as possible, as we may be able to grant a short extension.

Best wishes,
Dr Sasha Dall
mailto:proceedingsb@royalsociety.org

Associate Editor

Comments to Author:

Wonderful. The manuscript has been substantially improved and is really well written and presented. It is accessible and will be of significant interest (and useful) to a broad range of biologists. I was really pleased to see the section on reproducibility. Both expert reviewers are also very happy with the revised manuscript. They have suggested additional revisions, most of which are very minor. However there are two points identified by reviewer 2 that will require a bit of extra effort: the first is to check assumptions regarding the correlation structure for PGLS models, and the second is to check that the assumption of equal rates is appropriate in ancestral state reconstruction. In addition, there are minor editorial suggestions, to which I would like to add one suggestion regarding the abstract. I suggest reversing the logic/order of the sentences in lines 19 to 23 so that it clear that you are testing hypotheses regarding ecological drivers of eye size. Suggested wording (feel free to change/improve):

'We measured relative eye size and relative corneal size and tested whether they were predicted by six natural history traits hypothesised to be associated with the evolution of eye size. Anuran eye size was significantly correlated with habitat, with notable decreases in eye investment among fossorial, subfossorial, and aquatic species. Additionally, relative eye size was associated with mating habitat and activity pattern...'

I look forward to seeing this paper published!

Reviewer(s)' Comments to Author:

Referee: 3

Comments to the Author(s).

I have reviewed the manuscript entitled "Eye size and investment in frogs and toads correlate with adult habitat, activity pattern, and breeding ecology" and find it to be a very well written

and interesting study. It is a very important contribution to the literature, addressing a prominent gap in our understanding of species-level diversity patterns in vertebrates. I think it is an excellent paper, polished and without any major issues. Statistically, it is sound and extensive. The figures are beautiful, clear and easy to follow. The dataset is very impressive and commendable. I have only minor comments suggested for clarity.

The terminology behind allometry and allometric scaling as viewed from regression analyses could have some consideration:

“Developmental allometry” – usually called ontogenetic allometry

Hypoallometric – also known as negative allometry; could be good to clarify this at first use for readers used to that terminology

“high allometric slopes” – I think the authors are referring to the intercept differences here, and not the angle of the slope. Better to clarify, since it is ambiguous in places.

L66 (and 68): maybe change to say “eye size (absolute and relative to body size)” here, since “and allometric scaling” is ambiguous. Or make it clear that allometric scaling belongs to the eye size.

Methods: sampling – I don’t think that the sentences L105-108 needs to be a separate paragraph. This information can fit in after the museum collection sentences on L97 and into the measurements taken sentences after it.

L111 – “downloaded from dryad” unnecessary. Can simply have [31, 33] after phylogeny.

Fig. 3 – really beautiful figure. On the empty space, stylistic suggestion: could a schematic/silhouettes be put here showing eye sizes in two extreme frogs (ground dwelling vs scansorial)?

caption – please add in a brief explanation of what is mean of eye investment here. These two suggestions make the figure stand alone and would make an excellent graphical abstract.

I especially like the part of the predictions regarding eye size in adults where their larvae have a particular need for high visual acuity. I wonder if it is worth adding a sentence in the discussion clarifying that due to the adaptive decoupling hypothesis, eye size may not be a heritable trait across metamorphic boundary. Though to my knowledge of this literature, there are no papers that have looked at the heritability of eye size.

Finally, I’m glad to see the support for museum specimens highlighted here; the accuracy analysis of preserved and fresh material is not only an important finding, but also commendable and leading by example.

Referee: 1

Comments to the Author(s).

I am very happy with the modified version of this manuscript. The analyses are very detailed and the way how scripts were made available is exemplary. I have a few suggestions that should be addressed:

1) PGLS is currently implemented with a Brownian Motion correlation structure. Given that the data suggest that adaptive processes may shape evolutionary pattern, an underlying OU correlation structure may be more appropriate.

2) Ancestral state reconstruction are currently implemented with a single (equal) rate. One should check whether this is supported by the data.

I also would like to see more detail on the specific effect of eye size on visual performance, specifically with respect to the sensitivity of an eye to extended light sources vs point light sources. A larger aperture in absolute terms will result in better light sensitivity, while absolute eye size does not influence sensitivity to extended light sources (all else being equal).

- 3) As for cornea size, in how far can it be demonstrated that cornea size is correlated with maximum pupil diameter? Corneal diameter itself does not have a direct effect on visual performance. Corneal curvature would, however.
- 4) Lastly, I was wondering whether Ritland's 1982 dissertation on eye size across vertebrates should be cited. To the best of my knowledge Ritland's work was never published but the dissertation is available and has been cited in other comparative eye studies.

Author's Response to Decision Letter for (RSPB-2020-1393.R0)

See Appendix B.

Decision letter (RSPB-2020-1393.R1)

01-Sep-2020

Dear Dr Thomas

I am pleased to inform you that your manuscript entitled "Eye size and investment in frogs and toads correlate with adult habitat, activity pattern, and breeding ecology" has been accepted for publication in Proceedings B.

Open Access

Paper charges

All supplementary materials accompanying an accepted article will be treated as in their final form. They will be published alongside the paper on the journal website and posted on the online

figshare repository. Files on figshare will be made available approximately one week before the accompanying article so that the supplementary material can be attributed a unique DOI.

Sincerely,
Dr Sasha Dall
Editor, Proceedings B
mailto: proceedingsb@royalsociety.org

Associate Editor:

Comments to Author:

Thank-you for carefully considering and addressing the reviewer comments. I was very impressed by the care and effort that the authors put into revisions on both occasions. This paper will make a fine contribution to Proceedings B.

Appendix A

Response to reviewers:

Associate Editor

Comments to Author:

Your appeal has now been considered by the Editor. I am pleased to let you know that on this occasion, the Editor has decided to allow your appeal, and invites you to resubmit your manuscript to the journal. Specific comments from the Associate Editor are included below:

"The revision would not be trivial (as the appeal makes out) - it would need some careful thought and additional analyses."

Response: *We sincerely thank the editorial team for allowing us the opportunity to resubmit a revised version of our manuscript. We have reanalyzed our datasets based on the many helpful and constructive suggestions provided by the AE and reviewers. We have also substantially rewritten the manuscript as a result of these new analyses. We hope you will agree that these extensive revisions have led to a significant improvement to our manuscript and allow for more robust interpretations about the ecology and evolution of eye size in anuran amphibians within the broader context of vertebrate eye evolution.*

This study looks at eye size and investment in anurans, compares scaling relationships with other vertebrate groups and examines the relationship between eye investment and habitat. A strength of this study is the sample size and validation of the methods (comparison of museum and fresh specimens, reproducibility). However, analyses and results comprise a mix of phylogenetically corrected and uncorrected analyses. This makes results confusing and difficult to interpret because robust inferences can only be drawn from phylogenetically corrected analyses. I realise that phylogenetically uncorrected analyses were done to enable comparison across vertebrate groups and to previous studies, but phylogenetic information should be available for these groups, and with no phylogenetic correction, the comparison doesn't mean much.

Response: *We have redone all analyses so that now every scaling analysis is presented as phylogenetically corrected, and the manuscript is edited accordingly to center on these results. This includes phylogenetically corrected comparisons to other vertebrate groups which were lacking in our original submission because some of the published studies on other vertebrates did not include phylogenetic correction. In examining how absolute and relative eye sizes are distributed across ecological categories, we have decided to present both phylogenetically corrected and uncorrected analyses because our interpretations in the discussion relate to both evolutionary and functional implications of the patterns we observed. While phylogenetic correction is essential for evolutionary interpretations (e.g. positing that expanding into a new habitat may have resulted in the evolution of larger eye sizes) it is unnecessary for functional interpretation because absolute eye size has a direct, functional impact on vision, so regardless of whether two species have similarly large eyes because of common ancestry, they are still*

likely to see better than species with smaller eyes. Thus, we still include tests for differences in absolute eye size among ecological states as well as phylogenetically corrected relative eye sizes among states. In response to reviewer comments and suggestions, we now also include models that account for the phylogenetic distribution of eye size and ecology, with body size as a covariate, and these showed that our original findings were robust and yielded a new finding about activity period. We have clarified the text to outline our reasoning and the limitations of each analysis in terms of evolutionary vs. functional implications.

The biological significance of comparing the relationship between eye size and SVL in anurans, snakes and lizards is also unclear because of the difference in body shape (elongation).

Response: *The biological significance is derived from comparative allometry. Even if organisms have different intercepts due, for example, to differences in body shape/mass, comparing the slopes allows for interpreting differences/similarities in the interspecific growth trajectory of eyes with body size – which we think is biologically meaningful and highly relevant to our study. We have clarified this in the manuscript and mention differences in body plan (elongation in particular) in our interpretation of results. (Lines 278 – 281)*

Also, as I understand it, analyses relating eye investment to habitat were done using phylogenetically corrected relative eye size but tests for differences between habitats were not phylogenetically corrected. This means that phylogenetic non-independence in habitat was not accounted for, again making results difficult to interpret.

Response: *We have now included analyses that account for the phylogenetic non-independence of all factor variables (See Tables S8, S9, S10). Accounting for phylogenetic non-independence supported the significance of our original findings, and yielded a new finding about the correlation between activity period and eye size.*

Lastly, as reviewer 1 points out, the study would benefit from articulating clear hypotheses about relationships between eye investment and habitat. Because the biological significance of the results is unclear given the analyses, I cannot recommend acceptance in PRSLB. However, the reviewers have provided valuable feedback and I'm sure that this impressive dataset can be used to make a valuable contribution to our understanding of the evolution of eye investment in anurans.

Response: *We have added clearly articulated hypotheses and expectations to the introduction:*

“We propose that broad sampling of anuran phylogenetic and ecological diversity potentially relevant to vision may uncover correlations between anuran ecology and relative eye size. Because adult habitat and activity pattern are important drivers of relative eye size in other taxa, we predicted that (1) species active in fossorial (burrowing) and aquatic habitats would show reduced eye investment because they inhabit dark or highly attenuating environments,

and that (2) nocturnal species would invest in large eyes and/or large corneas to maximize sensitivity in scotopic (low-light) conditions. Additionally, because many anurans are most active during the breeding season and may be heavily reliant on vision at this time [27], we predicted that (3) species breeding in complex sensory habitats (e.g. on vegetation or near fast-flowing water) or (4) exhibiting sexual dichromatism (potentially related to conspecific signaling [28,29]) would invest in larger eyes for improved visual discrimination during breeding. Finally, because most anurans have a biphasic life history with decoupled larval and adult morphologies and ecologies [30], we predicted that species with (5) free-living larvae and (6) larvae active in terrestrial or lotic environments (where vision may be crucial to larval survival) would have larger adult eye sizes due to increased larval investment in vision." (Lines 70-84).

Reviewer(s)' Comments to Author:

Referee: 1

Comments to the Author(s)

I have read the manuscript on eye size in frogs with great interest. The visual system of frogs is indeed not well known, and any new data promise exciting findings. However, there are several shortcomings in the design of the study that question the validity of the results.

A major concern is the false equivalence of considering both traditional statistical methods and phylogenetically informed approaches. It has been demonstrated many times that one must account for phylogenetic covariance in the data when dealing with interspecific datasets. Why perform traditional analyses when a time-calibrated phylogeny is available?

Response: *We originally presented both traditional allometric methods alongside phylogenetically informed approaches in order to make our work comparable to the broadest possible collection of published literature. We agree that a consideration of phylogeny is important, so we have reanalyzed all vertebrate eye size data compiled from published literature so that they are phylogenetically corrected. Accounting for phylogeny in the scaling relationships of all vertebrate groups examined had very little effect on the model fits. Furthermore, as mentioned above in our responses to the AE there are direct functional interpretations that are valuable prior to phylogenetic correction for some analyses; this is why we also still retain these analyses in the revised manuscript. However, we have minimized discussion of non-phylogenetic methods in our revised draft.*

All SMA analyses and results should be considered invalid (both for frogs and across vertebrates). The potentially interesting finding of an isometric slope of eye growth compared to body size, which could have been a major result of the paper, is not supported: the possible

isometry is advertised in the abstract, but the phylogenetically informed results demonstrate that isometry is rejected.

Response: *We have removed the statement about isometry from the abstract. We originally included this discussion to compare anuran eye scaling with published studies of other vertebrate groups, some of which did not include phylogenetic correction. However, we have re-done all analyses to include phylogenetic information for all vertebrate groups included in the paper, so we have now addressed this critique.*

In addition, the choice of phylogenetic comparative methods is inadequate. There are many modern comparative approaches that allow tests for adaptive evolution in a modeling framework, routinely employed in many papers on trait evolution (OUwie, mvMORPH, l1ou, bayou, and many other R packages). A thorough analysis of eye evolution in frogs should go far beyond a PGLS (which would need revision to be a true phylogenetic ANCOVA).

Response: *We are aware there is a wide range of evolutionary models available for studying adaptive evolution; however, we do not agree that they are appropriate given the questions and aims of our study. We believe our revised methods are sufficient to address our questions. The aims of our study were to 1) determine the scaling relationship between eye size and body size in anurans and compare that to scaling in other vertebrates, and 2) test a suite of ecological variables relevant to visual ecology for potential correlations to absolute and relative eye size in anurans. As this is the first dataset of its kind, we argue that we have met the aims of our study and that further questions exploring in-depth adaptive evolution of anuran eyes through complex evolutionary modeling are beyond the scope of the present study. Upon the publication of our paper, which includes our full raw morphological dataset, ecological dataset, and meticulously annotated scripts to repeat all of our analyses, it will be possible for someone interested in deeper evolutionary questions to build on our work and address these questions in the future as the reviewer suggested.*

There are additional concerns regarding the data collection and study design. To enable large-scale comparisons across different clades some eye diameter data were converted into length data which I would think is problematic because not all eyes have the same ellipticity.

Response: *In comparisons across vertebrates, some papers used the axial length of the eye, which cannot be measured non-destructively in museum specimens. This is why we included a dataset on fresh, field-caught anurans that we dissected whole eyes from and directly measured for axial length and transverse diameter. The reviewer is correct that not all eyes have the same ellipticity, however, we found that in our fresh dataset axial length was highly correlated with transverse diameter ($R^2 = 0.96$); thus, our transformation of data from transverse diameter to axial length is statistically supported and reasonable to facilitate broad comparisons across vertebrate groups. We have further highlighted this in the revised methods section for clarity. (Line 191, Figure S2A)*

Also, there is one comparison where snakes, geckos, and frogs are compared by putting SVL length on the X-axis. That comparison does not make much sense because snakes are so elongated, and even geckos are much more elongated than frogs are.

Response: *We disagree that body shape variation invalidates a comparison of eye size compared to body size. Yes, snakes are much more elongate than frogs, but they invest less in eye tissue per body length than frogs. Importantly, the scaling relationships between eye size and body size are not simply about the intercepts, but also about the slopes, which interestingly are similar across snakes, geckos and anurans despite the great differences in body shape. In response to the reviewer's comments, we have clarified this in the results section. (Lines 278-281)*

Finally, the questions of the manuscripts are not guided by clear hypotheses. What are the exact implications of eye size on vision and what eye size changes are expected because of habitats? An obvious pathway may be a test of MacIver's 2017 buena vista hypothesis (eye size changes between water and air/land).

Response: *We have addressed this comment with additional text in the introduction stating our specific hypotheses, as noted above in our response to the AE.*

Referee: 2

Comments to the Author(s)

This manuscript, entitled "Eye size and investment in frogs and toads correlate with adult habitat and breeding ecology", aims to unravel the correlation between relative eye size and different ecological and behavioural traits. The authors collected several morphological measurements and ecological and behavioural categorical traits from 220 species comprising 55 anuran families. They also compare anuran relative eye sizes with other vertebrate groups. They found that relative eye size is significantly correlated (without phylogenetic correction) with habitat, and that eye size is proportionally inverse to fossoriality degree. They also found a significant correlation between relative eye size and mating habitat and sexual dichromatism. Finally, the authors found that frogs have relatively large eyes compared to other vertebrates.

Overall, I quite enjoyed reading this manuscript. It is well written, the authors have performed a lot of sound analyses, and it tackles a very important and exciting question on morphological adaptation. Therefore, I think this will be a very interesting contribution and I recommend it to be published in Proceedings B.

Response: *We thank the reviewer for the encouraging assessment and constructive criticism.*

However, I have a few suggestions, particularly around the use of phylogenetic corrections in the analyses (and subsequent discussion of the results). Incorporating the phylogeny should be a selling point of this paper, since previous studies did not do it. However, the authors mostly focused on the non-phylogenetically corrected analyses (and results interpretation).

Response: *We have shifted the focus of the manuscript onto the phylogenetically corrected results, as well as incorporated phylogenetic correction into our comparison of anurans to >1200 species from other vertebrate groups. We do retain the relevant results from some non-phylogenetically corrected analyses, as described above.*

As well as that, the authors should make it clearer that fish have similar relative eye size ‘trend’ as fish. In Table S6 the authors present SM fits for eye scaling in different vertebrate groups, and the slope 95% C.I. of anurans and fish overlap. I think they should develop this in the discussion and change ‘vertebrate’ for ‘tetrapod’.

Response: *We appreciate the reviewer’s comment and have revised the text of the results (Lines 276 – 278) and the discussion (Lines 350 -352) to highlight this similarity between fishes and anurans. We found it easiest to refer to ‘amniotes’ to describe the similarity between fish and frogs vs. the other groups assessed, as ‘tetrapods’ includes amphibians but excludes fish.*

I have provided a few comments that I hope will help to improve this paper. However, some of the comments I am giving are a personal preference, so while I hope they will help to improve the paper, I do not expect the authors to perform or reply to all of them.

Response: *We thank the reviewer for their section specific comments and have used them to greatly enhance the revised manuscript.*

ABSTRACT:

19 – the authors should clarify whether they are referring to non-phylogenetically corrected results or not. It is unclear here.

Response: *We acknowledge that our original submission was not always clear when referring to PGLS or SMA results. In the revised paper we are more explicit in what analyses we are referring to and focus on PGLS results. We hope this addresses the reviewer’s concern.*

27 – Most of the results focus on non-phylogenetically corrected data, so it would not be suitable to infer and interpret macroevolutionary patterns.

Response: *We now use only phylogenetically corrected analyses to address macroecological patterns. We restrict the use of uncorrected analyses to addressing functional patterns.*

INTRODUCTION:

36 - Reference needed

Response: *Reference has been added.*

41-49 – This paragraph provides important technical background information, but it isn't explained as well as it could be. The authors might want to consider re-writing it.

Response: *We have edited this paragraph to streamline and hopefully clarify the text.*

43 – Reference needed

Response: *Reference has been added.*

45 – Is this a comparison to a camera? I think it needs a little bit more clarification

Response: *We have replaced the comparison to a camera to clarify this.*

79 – It might be good re-phrasing this sentence in a more positive way, Also, could the authors add other differences to that paper? I think this manuscript will be a very important new contribution, but here it sounds a bit like the main difference between their manuscript and the paper they cite is 'adding extra sampling'.

Response: *To address this and another reviewer comment, we have altered this paragraph to be more explicit about why we investigated the ecological variables we chose to (most of which differed from the previous study we cite), and provide explicit hypotheses for how eye size may correlate with each variable. We have also moved more detailed discussion of the methods and results of our comparison the previous study to the Supplementary Materials so that we could explain our comparisons in detail and focus on our own hypotheses and methods in the main text.*

METHODS:

87 - The authors should define RM, AL and AD in this section.

Response: *We have now defined all measurement acronyms in the sampling and measurement section of the methods.*

100 – It might be better to talk about sampling sizes, etc. earlier in the M&M section.

Response: *We have moved discussion of sample sizes to the measurement section of the methods. (Lines 93-95)*

118 - Please refer to a specific table.

Response: *We have now referred to the appropriate supplemental data.*

122 – It would be useful for the readers if the authors explain why they used SMA instead of ordinary least squares (OLS). It might be good to mention degree of asymmetry, bias in the slope estimates of SMA vs. OLS, etc.

Response: *We have now included both OLS and SMA analyses and while they don't differ much, we present them both for comprehensiveness. However, in response to other suggestions we focus mainly on PGLS results throughout the paper.*

154 – The authors need to define AL. I was not able to find it anywhere in the methods.

Response: *We have corrected this oversight.*

156 - Define AD.

Response: *We meant AL and have corrected this oversight.*

174 – Provide more details.

Response: *The statement made in Line 174 of the original submission was: "We then tested whether EDs, eye investments, or corneal investments differed among states for each of the six ecological traits (Table S1). Our data were not normally distributed, so we used nonparametric Kruskal-Wallis tests to convert values to ranks and then test for equal distribution across states."*

We were unable to understand what level of detail the Reviewer was requesting as this statement seems (to us) to be quite descriptive. However, in our revision of the manuscript this section of the methods has been edited and hopefully will now be clearer.

DISCUSSION

310-313 – See my main comment regarding phylogenetic correction. The phylogenetically-corrected result is rushed in this paragraph. I think it should be discussed further. As well as that, the last sentence sounds a bit like they are not doing it because previous studies did not use phylogenetic correction.

Response: *In the revised manuscript we have addressed this issue by adding phylogenetically-corrected analyses for all allometric comparisons. Uncorrected scaling comparisons are no longer discussed in the paper.*

318 – which states?

Response: *We outline the categorical states for each ecological trait in the Methods (Lines 125-130) and in Table S1.*

319 “with” body size instead of “to”

Response: *We have corrected this in the revision.*

328 – “where given body mass” – strange wording

Response: *This was awkward wording and we have rephrased.*

337 – And scansorial?

Response: *We have changed this sentence for clarity: “Interestingly, mean eye investment increased from fully fossorial (0.65x) to subfossorial (0.91x) to non-fossorial (ground-dwelling, semiaquatic, scansorial) anurans (1.12x – 1.24x)...” (Lines 306-308)*

356-358 – This needs clarification

Response: *We have clarified these two sentences with the following restructuring: “Our study revealed that anurans have large relative eye sizes compared to other vertebrates (Figure 5). For a given body mass, anurans had similar axial lengths to birds (Figure 5A), which previously have shown the highest investments in eye size among extant vertebrates [49].” (Lines 348-351)*

395 – I think it is important to further explain this bit (“... difference in how we categorized states”)

Response: *We have clarified this with the following revision: “Our novel finding that relative eye investment correlates with adult habitat and activity pattern was different from previously reported results [26], likely due to increased sampling and the inclusion of broader ecological diversity (e.g. fossorial species, diurnal species) in our analyses” (Lines 376-379). We also include details of how we compared our categorization of habitat to the previously published study that found no correlation between eye size and habitat in the Supplemental Methods and Supplemental Results.*

FIGURES, TABLES, AND SUPPLEMENTARY MATERIAL

The authors might want to consider changing the colour scheme used. I struggled to see each of the categories in some of the plots.

Response: *We have decreased the transparency of point colors in anuran scaling and ecology plots (Figures 2 - 4) to increase contrast and more easily distinguish between groups, and have labeled the x-axis categories in Figure 3B & C. We have also changed the color palettes for Figure 5 and decreased the transparency of points. Finally, we have broken Figure 5 into its component taxa and plotted separately in Figure S3 so that it is easier to see individual datasets, sources, and scaling relationships.*

Fig. 2 – Be consistent (use RM instead of body mass. Similarly ED instead of eye size.

Response: *We have corrected this in the captions.*

Fig. 4, Fig. S4-S9 – Since the taxonomic sampling presented here is uneven across the anuran (complete phylogeny), this will greatly affect the ancestral state reconstruction. I think the authors should point out this bias.

Response: *We have now pointed this out in the methods section of the paper when we discuss ancestral state estimation for figures. (Lines 183-184)*

Appendix B

Response to reviewers:

Associate Editor

Comments to Author:

Wonderful. The manuscript has been substantially improved and is really well written and presented. It is accessible and will be of significant interest (and useful) to a broad range of biologists. I was really pleased to see the section on reproducibility. Both expert reviewers are also very happy with the revised manuscript. They have suggested additional revisions, most of which are very minor. However there are two points identified by reviewer 2 that will require a bit of extra effort: the first is to check assumptions regarding the correlation structure for PGLS models, and the second is to check that the assumption of equal rates is appropriate in ancestral state reconstruction.

In addition, there are minor editorial suggestions, to which I would like to add one suggestion regarding the abstract. I suggest reversing the logic/order of the sentences in lines 19 to 23 so that it clear that you are testing hypotheses regarding ecological drivers of eye size. Suggested wording (feel free to change/improve):

'We measured relative eye size and relative corneal size and tested whether they were predicted by six natural history traits hypothesized to be associated with the evolution of eye size. Anuran eye size was significantly correlated with habitat, with notable decreases in eye investment among fossorial, subfossorial, and aquatic species. Additionally, relative eye size was associated with mating habitat and activity pattern...'

I look forward to seeing this paper published!

Response: *We are grateful to the AE and reviewers for volunteering their time and insights to improve the manuscript. We have addressed each reviewer suggestion separately below. In response to the AE's suggestion about the abstract, we have edited lines 18 to 23 to read: "We measured relative investment in eye size and corneal size for 220 species of anurans representing all 55 currently recognized families and tested whether they were correlated with six natural history traits hypothesized to be associated with the evolution of eye size. Anuran eye size was significantly correlated with habitat, with notable decreases in eye investment among fossorial, subfossorial, and aquatic species. Relative eye size was also associated with mating habitat and activity pattern."*

Reviewer(s)' Comments to Author:

Referee: 3

Comments to the Author(s).

I have reviewed the manuscript entitled “Eye size and investment in frogs and toads correlate with adult habitat, activity pattern, and breeding ecology” and find it to be a very well written and interesting study. It is a very important contribution to the literature, addressing a prominent gap in our understanding of species-level diversity patterns in vertebrates. I think it is an excellent paper, polished and without any major issues. Statistically, it is sound and extensive. The figures are beautiful, clear and easy to follow. The dataset is very impressive and commendable. I have only minor comments suggested for clarity.

The terminology behind allometry and allometric scaling as viewed from regression analyses could have some consideration:

“Developmental allometry” – usually called ontogenetic allometry

Response: *We have changed the phrase “developmental allometry” to “ontogenetic allometry” throughout the manuscript (Lines 389 & 394)*

Hypoallometric – also known as negative allometry; could be good to clarify this at first use for readers used to that terminology

Response: *We have added this clarification into the first mention of hypoallometry (Line 233: “There was hypoallometric (slope < 1, negative allometry) interspecific scaling between anuran ED and RM...”)*

“high allometric slopes” – I think the authors are referring to the intercept differences here, and not the angle of the slope. Better to clarify, since it is ambiguous in places.

Response: *When we mentioned “high allometric slopes,” we were intentionally referring to the slopes (the change in eye size per unit change in body mass across species). Anurans do have large relative eye sizes (which is tied to intercept) compared to other vertebrates, but they also have a higher/steeper slope than many groups, so that large-bodied frogs still have large relative eye sizes. To clarify our meaning here, we have changed Line 280 to read “anurans have large relative eye sizes and steep allometric slopes”, and changed subsequent mentions of slope to “steep(er)” rather than “high(er)” (Lines 282, 284, 371).*

L66 (and 68): maybe change to say “eye size (absolute and relative to body size)” here, since “and allometric scaling” is ambiguous. Or make it clear that allometric scaling belongs to the eye size.

Response: *We have revised Lines 74-75 (“Despite this, anuran eye size (absolute and relative to body size) is largely unstudied outside of a few families” and Line 77 (“eye size and eye-body allometry in frogs and toads have not previously been compared to other vertebrate groups”) to reflect this suggestion.*

Methods: sampling – I don’t think that the sentences L105-108 needs to be a separate

paragraph. This information can fit in after the museum collection sentences on L97 and into the measurements taken sentences after it.

Response: *We have integrated the information on fresh specimens into the paragraphs on preserved specimens (Lines 103-119) as suggested by the reviewer.*

L111 – “downloaded from dryad” unnecessary. Can simply have [31, 33] after phylogeny.

Response: *We have deleted the unnecessary text from Line 104 as suggested by the reviewer.*

Fig. 3 – really beautiful figure. On the empty space, stylistic suggestion: could a schematic/silhouettes be put here showing eye sizes in two extreme frogs (ground dwelling vs scansorial)?

caption – please add in a brief explanation of what is mean of eye investment here. These two suggestions make the figure stand alone and would make an excellent graphical abstract.

Response: *We have added silhouettes for species showing different relative eye investments to the blank spot in Figure 3 (now panel D), and thank the reviewer for this great suggestion. We have also added the following explanation of relative eye investment to the figure caption: “Eye investment represents the species residuals from the PGLS fit for log-transformed ED vs. the cube root of body mass (RM), exponentiated so that species that fall precisely along the allometric fit have an eye investment of 1x the value predicted by fit, those that fall below have eye investments of <1x (small relative eye sizes), and those that fall above the fit have eye investments of >1x (large relative eye sizes).” Finally, for the new panel, we have added the following text to the caption: “(D) Silhouettes of four species (colored by adult habitat) that exhibit differences in eye investment relative to body mass (in bold).*

I especially like the part of the predictions regarding eye size in adults where their larvae have a particular need for high visual acuity. I wonder if it is worth adding a sentence in the discussion clarifying that due to the adaptive decoupling hypothesis, eye size may not be a heritable trait across metamorphic boundary. Though to my knowledge of this literature, there are no papers that have looked at the heritability of eye size.

Response: *We have added the following text to the discussion (Lines 390-391): “In biphasic anurans this is especially intriguing, as morphological evolution can be decoupled for larvae and adults [80]” and cited Valero et al. (2017) – “Transcriptomic and macroevolutionary evidence for phenotypic uncoupling between frog life history phases”.*

Finally, I’m glad to see the support for museum specimens highlighted here; the accuracy analysis of preserved and fresh material is not only an important finding, but also commendable and leading by example.

Response: *We thank the reviewer for their thoughtful and constructive feedback on the manuscript.*

Referee: 1

Comments to the Author(s).

I am very happy with the modified version of this manuscript. The analyses are very detailed and the way how scripts were made available is exemplary. I have a few suggestions that should be addressed:

1) PGLS is currently implemented with a Brownian Motion correlation structure. Given that the data suggest that adaptive processes may shape evolutionary pattern, an underlying OU correlation structure may be more appropriate.

Response: *Morphological traits are known to show high levels of phylogenetic correlation. Our analyses employed the `caper` package in R to implement PGLS, which is based on the methods presented in Freckleton et al. (2002) – “Phylogenetic analysis and comparative data: A test and review of evidence”. This method uses maximum likelihood to estimate Pagel’s lambda, and uses this estimate of lambda to make the correlation structure for the PGLS. If the ML estimate for Pagel’s lambda = 1, then the model employs a Brownian motion correlation structure, and if Pagel’s lambda = 0, then the model uses a correlation structure that assumes independence across species (no phylogenetic correlation) and the model does not differ from a non-phylogenetic GLS model.*

In general, a maximum likelihood approach to estimating Pagel’s lambda is broadly used in macroecological and evolutionary studies similar to ours, and we feel that it is the most conservative and appropriate model to test our specific hypotheses. Using BM vs. OU models should not have a major impact on the results of our statistical tests, and OU models introduce additional parameters to fit (e.g. alpha), are more sensitive to measurement error, and are at increased risk of Type 1 error (see Cooper et al. [2015] – “A cautionary note on the use of Ornstein Uhlenbeck models in macroevolutionary studies”).

However, to follow up on this reviewer suggestion, we have compared models by AIC scores for phylogenetic GLS models using 1) pure Brownian motion, 2) ML for OU, and 3) ML for Pagel’s lambda. We did this for our models of eye-body allometry and for each of the PGLS models in our study that showed a correlation between eye size and ecology (adult habitat, mating habitat, and activity pattern). In all cases, the GLS model using the ML estimate of lambda was significantly better ($\Delta AIC > 2$) than the models using Brownian motion or OU. Thus, we are confident that our original analyses using a ML estimation of lambda are appropriate for our dataset.

2) Ancestral state reconstruction are currently implemented with a single (equal) rate. One should check whether this is supported by the data.

Response: *We originally included ancestral state estimations on Figure 3A and Supplemental Figures S4 – S9 for each of the ecological traits we coded to highlight the many ecological transitions that have likely occurred given the phylogenetic distribution of extant ecotypes. However, as the reviewer points out here, ancestral state estimation for discrete character evolution requires assumptions about transition rates between states. Additionally, models are more reliable when there is information about deeper nodes, for example, from fossil evidence. Finally, some of our traits have more than two states (e.g. habitat has 6 states), and we cannot reasonably assume that the likelihood of transition is equal in all directions across all states, but do not have data to improve these model assumptions. Given these issues, we have decided to remove the ancestral state estimations from our figures, and have edited Figures 3A and S4-S9 to reflect this minor change. We have also deleted the following sentence from the methods section (originally Line 201): “We determined the probability distribution for ancestral ecological states based on a single-rate model of discrete character evolution (though note that ancestral state estimation is highly sensitive to taxon sampling and tree topology [47])”.*

I also would like to see more detail on the specific effect of eye size on visual performance, specifically with respect to the sensitivity of an eye to extended light sources vs point light sources. A larger aperture in absolute terms will result in better light sensitivity, while absolute eye size does not influence sensitivity to extended light sources (all else being equal).

Response: *We have added the following text to the first two paragraphs of the introduction (Lines 37-47) to better explain the tradeoffs between sensitivity and resolution, and why having a larger eye can improve one and/or the other:*

“Sensitivity increases when each retinal photoreceptor views a larger solid angle of the visual scene, allowing more photons to reach each detector. Resolution, however, increases when the solid angle sampled by each photoreceptor is decreased, dividing the external visual scene into finer detail [3]. Thus, an improvement in one aspect of vision comes at the cost of another, unless the size of the eye is increased.

When an eye is scaled up with constant proportions, acuity increases, while sensitivity to extended visual scenes does not change. This is because acuity is proportional to focal length, while sensitivity is proportional to the ratio of aperture to focal length [2,4]. However, a number of morphological and neural strategies can improve sensitivity at the cost of acuity [5], so a larger eye can improve both sensitivity and/or resolution compared to a smaller eye.”

We agree with the reviewer that our previous introduction to sensitivity was too vague. In the new text, we have clarified how optical sensitivity is related to eye dimensions (aperture and focal length) for viewing extended scenes. We think that this is the most broadly relevant visual task for our study system, and for the sake of space have not gone into detail about how sensitivity differs for viewing point sources (where the image falls onto a single photoreceptor,

as with stars or with flashes of distant bioluminescence in the deep sea). However, we have now explicitly said that we are discussing sensitivity for extended scenes to clarify this point, and have cited works that further delve into the details of vision for extended scenes vs. for point sources.

3) As for cornea size, in how far can it be demonstrated that cornea size is correlated with maximum pupil diameter? Corneal diameter itself does not have a direct effect on visual performance. Corneal curvature would, however.

Response: *We recognize that cornea diameter does not have a direct effect on vision; our hypotheses about corneal size are related only to its putative correlation to the maximum potential diameter of the pupil (aperture), as it is the ratio of aperture to focal length that determines sensitivity (we have also clarified this relationship better in the paper in response to the reviewer's previous suggestion). Because pupil diameter is dependent on light environment and muscular contractions at the time of collection and preservation, we were unable to measure maximum pupil diameter in preserved museum specimens or in the fresh, field-caught specimens we used in the study, so we have used corneal diameter as a proxy. Generally, we would not expect the pupil diameter to exceed the corneal diameter, and anecdotally we have not seen this in any of the thousands of live anurans we have cumulatively seen. The pupil does frequently expand to fill a majority of the cornea, and thus cornea diameter likely represents the upper limit of possible aperture diameter. We note that a published study on eye dimensions in lizards used cornea diameter as a proxy for aperture and found that, as predicted, scotopic species had larger cornea diameters relative to axial lengths of the eyes (Hall [2008] – "Comparative analysis of the size and shape of the lizard eye"). To clarify this in the text, we have made the following changes:*

In lines 63-64 we have changed "large corneas" to "large apertures (pupils) relative to focal lengths" to more accurately describe the pattern that cornea is approximating in our study. ("Activity pattern can also influence eye size, and nocturnal animals often have relatively large eyes and/or eyes with large apertures (pupils) relative to focal lengths to increase sensitivity in low-light conditions").

In outlining our specific predictions, we have also now explicitly mentioned that corneal diameter is a proxy for the maximum possible pupil diameter in our study (Lines 85-87: "(2) nocturnal species would invest in large eyes and/or large corneas (approximating maximum pupil diameter) to maximize sensitivity in scotopic (low-light) conditions.")

4) Lastly, I was wondering whether Ritland's 1982 dissertation on eye size across vertebrates should be cited. To the best of my knowledge Ritland's work was never published but the dissertation is available and has been cited in other comparative eye studies.

Response: *We have now cited Ritland (1982) in Lines 77-79 ("eye size and eye-body allometry in frogs and toads have not previously been compared to other vertebrate groups in a*

phylogenetic framework (though see [29])” and in Lines 361-362 (“Relative eye size is often used as an indirect measure of the importance of vision to a taxon (e.g., [29,66,79])”.

We thank the reviewer for their insights and careful thought about the manuscript.